# The Evolution of Photocatalytic Membrane Reactors over the Last 20 Years: A State of the Art Perspective

**Raffaele Molinari \***[ID]**, Cristina Lavorato and Pietro Argurio**[ID]

Department of Environmental Engineering, University of Calabria, Via P. Bucci, Cubo 45/A,
I-87036 Arcavacata di Rende, CS, Italy; cristina.lavorato@unical.it (C.L.); pietro.argurio@unical.it (P.A.)
\* Correspondence: raffaele.molinari@unical.it; Tel.: +39-0984-496699

**Abstract:** The research on photocatalytic membrane reactors (PMRs) started around the year 2000 with the study of wastewater treatment by degradation reactions of recalcitrant organic pollutants, and since then the evolution of our scientific knowledge has increased significantly, broadening interest in reactions such as the synthesis of organic chemicals. In this paper, we focus on some initial problems and how they have been solved/reduced over time to improve the performance of processes in PMRs. Some know-how gained during these last two decades of research concerns decreasing/avoiding the degradation of the polymeric membranes, improving photocatalyst reuse, decreasing membrane fouling, enhancing visible light photocatalysts, and improving selectivity towards the reaction product(s) in synthesis reactions (partial oxidation and reduction). All these aspects are discussed in detail in this review. This technology seems quite mature in the case of water and wastewater treatment using submerged photocatalytic membrane reactors (SPMRs), while for applications concerning synthesis reactions, additional knowledge is required.

**Keywords:** photocatalytic membrane reactor; photocatalysis; wastewater treatment; degradation of recalcitrant pollutants; organic synthesis; photocatalytic reduction; photocatalytic oxidation

## 1. Introduction

Heterogeneous photocatalysis (HPC) is an advanced oxidation process (AOP) based on the use of ultraviolet (UV), visible (VIS) or infrared (IR) radiation in the presence of a semiconductor (the photocatalyst) to generate an oxidizing/reducing species [1,2].

The main difference between HPC and conventional catalysis consists in the photonic activation mode of the catalyst, which replaces the thermal activation that is characteristic of conventional catalysis. The photonic activation of the photocatalyst can be done thanks to the presence in the electronic structure of the photocatalyst (i.e., a semiconductor) of a valence band (VB) and a conduction band (CB), which are separated by a band gap of energy $(E_g)$ [3,4]. The excitation of the photocatalyst happens when photons, with energy $(h\nu)$ at least equal to the $E_g$, impact on the photocatalyst surface, thus promoting electrons $(e^-)$ from the VB to the CB. Then, a hole $(h^+)$ is formed in the VB [2,5]. The so-formed electron/hole couples promote both reduction and oxidation of the adsorbed substrate, generally by means of a mechanism involving the formation of radical species [6].

HPC has been broadly studied since 1972, when Fujishima and Honda [7] achieved water splitting using UV light and a $TiO_2$ photoanode in combination with a Pt counter electrode immersed in an aqueous electrolytic solution. Since then, HPC has been the subject of a great number of experiments, largely related to remediation processes such as water and wastewater treatment, where organic and inorganic pollutants are totally degraded into innocuous substances [8–13]. This is obtained because high-energy UV-light frequently induces highly unselective reactions, which happen by radical mechanisms [14,15]. Nevertheless, in recent years, studies have also been performed on the application of HPC for synthetic purposes [2,16–18], such as selective reduction [19–23] and oxidation [24–28], characterized by less by-product formation and the use of visible light as an energy source.

HPC has many advantages over concurrent traditional processes both in environmental remediation and synthetic usage, making it a sustainable green approach [29–35]. First, it works under mild conditions (usually ambient temperature and pressure). Additionally, HPC permits the use of greener and safer catalysts (mainly $TiO_2$) in contrast to the more dangerous heavy metal catalysts usually employed in thermally induced catalysis [34,35]. Moreover, HPC requires the use of mild oxidizing species (e.g., molecular oxygen) and permits the mineralization of refractory and non-biodegradable contaminants with the formation of innocuous by-products. HPC is characterized by high versatility, since it can be applied to a wide range of substrates in liquid, solid, and gaseous phases, even if diluted, and it requires very few auxiliary additives. Last, but not least, it allows for the possibility of using renewable solar energy, and HPC can be coupled with other physical and chemical technologies [36].

Despite the abovementioned advantages, the application of HPC at an industrial scale is still limited, mainly because of the costs related to the recovery and reuse of the heterogeneous photocatalyst and of the poor process selectivity [1,37].

To overcome the difficulties related to the separation of the photocatalyst, the coupling of HPC with membrane separation (MS) processes was proposed around the year 2000. An MS process is a physical technique that does not involve a phase change, permitting us to achieve the required separation by operating in continuous mode. By synergistically coupling MS with HPC, photocatalytic membrane reactors (PMRs) are a very promising technology since they permit us to achieve the minimization of environmental and economic impacts [38–41]. The advantages of PMRs mainly lie in the possibility of continuously operating systems in which chemical reactions, photocatalyst recovery and reuse, separation of chemicals from the treated effluent, and/or recovery of the products simultaneously occur. Therefore, the use of PMRs is a promising approach in view of the large-scale application of HPC. Improvements to process efficiency obtained in PMRs compared to conventional photoreactors, modularity, and easy scale-up have led to a significant growth in interest in the area of PMRs over recent years [42]. Nevertheless, the full-scale application of PMRs still requires exhaustive investigations to enhance their performance.

The design of a photocatalytic reactor (PR) is very difficult, since photocatalysis is an intrinsically heterogeneous process where the three components involved in the photocatalytic process are in different phases. The first component is the substrate, the second one is the solid photocatalyst, and the third one is the light photons, massless particles that promote the process by exciting the photocatalyst. Considering that in a PMR there is also the MS to be coupled with the HPC process, it is clear to see that the design of a PMR requires a highly inter-disciplinary approach involving the knowledge of chemical, mechanical, and environmental engineering concepts [43]. Considering these aspects, two main configurations of PMRs can be distinguished: a first one, called slurry PMRs, where the photocatalyst is suspended into the reaction environment, and a second PMR configuration where the photocatalyst is immobilized on a substrate material acting as a membrane (a photocatalytic membrane). It should be immediately pointed out that there is no configuration of PMR that is suitable for all applications, but that the different PMR configurations present specific advantages and disadvantages depending on the specific application [1,5,44–47]. Considering the coupling of HPC and MS processes, the development of the PMR with a suspended photocatalyst led to the development of systems in which the two processes take place in the same unique apparatus (the so-called integrative-type PMRs) [48,49], and systems where the HPC and MS processes take place in two separate apparatuses (the so-called split-type PMRs). In the latter, the photoreactor and the membrane separation unit are adequately integrated [29,50,51]. PMRs with an immobilized photocatalyst, named as PMRs with photocatalytic membrane (PM) [52], are intrinsically integrative-type. In those systems, it is important to have membranes resistant to irradiation, thus avoiding membrane photodegradation during the photocatalytic run. Moreover, based on the different position of the light source, different PMR configurations

can be distinguished. In particular, the irradiation source can be: (i) above/inside the vessel containing the feed solution; (ii) above/inside the cell containing the membrane; and, (iii) above/inside an additional vessel that can be located between the feed tank and the cell containing the membrane [45].

Regarding the MS process to be coupled with the HPC process, it should be pointed out that no MS exists that is best for all applications. Different MS can be coupled with the HPC process with the aim of increasing the PMR performance, depending on the particular application. Different PMRs, obtained by coupling HPC with MS (e.g., ultrafiltration (UF), nanofiltration (NF) [53–56], membrane distillation [57,58], membrane dialysis [59,60], and pervaporation [61–64]) have been employed in water treatment for degradation of various organic pollutants (such as dyes, pharmaceutically active compounds, and other pollutants) and in the synthesis of organics (such as phenol, vanillin, and phenylethanol).

In parallel to this evolution regarding PMR design and type of application, research in recent years has intensified regarding the development of photocatalysts with high photocatalytic activity under near UV and visible light [65]. The interest in using renewable energy sources, such as wind or solar light, has increased significantly in recent years due to increased energy requirement and pollution issues [39]. Most of the traditional photocatalysts are activated only by UV light energy, representing a small fraction (about 5%) of the solar spectrum. Thus, the remaining 95% (45% of visible light and 50% of near-infrared (NIR)) of the solar spectrum remains unutilized. Therefore, much work is needed to extend the light response from UV to visible light to increase the efficiency of solar energy [21,66–69]. Regarding these aspects, recent research has been concentrated on improving the full solar spectrum harvesting capacity by doping the photocatalyst with noble and transition metals, introducing two or more metal ions as co-dopants, forming heterostructures by combining different band gap materials, and controlling photocatalyst morphology [70].

The development of different photocatalysts demonstrates that the prepared materials are characterized by different responses to light irradiation depending on the irradiation range of interest. Based on this, various types of light sources are available, such as xenon [71,72], mercury [73], and deuterium lamps. In recent years, laser induced photocatalysis has also been studied. Among different types of lamps, LED lamps that emit in the UV or UV-VIS range are garnering interest because they are characterized by promising efficiency and have the possibility of being powered by photovoltaic panels [74,75].

In this review, the evolution of the scientific knowledge on photocatalytic membrane reactors is reported. It describes how some initial problems (membrane material, membrane configuration, membrane process, coupling membrane-photocatalyst, type of photocatalyst, irradiation source, and irradiation mode of the photocatalyst) have been solved/reduced over time to reach, in the near future, the ability to transfer the results from the laboratory to the industrial scale. The scientific community working in this field, and young researchers in particular, can benefit from this work that will hopefully aid in generating new ideas.

## 2. General Trend of Evolution of PMRs in the Years from 2000–2020

As reported in the Introduction, the idea of working on the advantages of HPC and solving/diminishing the impact of its drawbacks was introduced around the year 2000 by its coupling with MS. In particular, when searching the term "photocatalytic membrane reactor" both in the WOS and in the Scopus database, the earliest work reported is dated 1993 [76]. In this work, the application of an annular photoreactor for the photocatalytic degradation of aqueous solutions of formic acid and atrazine over a titanium dioxide ceramic membrane was studied. The proposed system, in which the photocatalyst was entrapped in the ceramic membrane, achieved small degradation rates. Thus, a recycle of the permeate was used to increase system performance. Despite that, the experimental results showed a very low quantum efficiency for atrazine photodegradation.

In 1994, in the ASME-JSES-JSME International Solar Energy Conference, Enzweiler et al. [77] reported the results obtained in their photocatalytic destruction of benzene,

toluene, ethylbenzene, and xylene in contaminated ground water over Degussa P25 $TiO_2$ in a pilot scale photocatalytic reactor using solar radiation and artificial UV light in the presence of hydrogen peroxide. In this work the membrane filtration technology was proposed and tested for catalyst recovery. The authors demonstrated that a slurry photocatalyst was about three times faster than an immobilized photocatalyst under the same conditions, and that the membrane separation made it possible to obtain the complete recovery and recycle of the catalyst with relatively low energy costs.

After these two pioneering studies, in 1999 (i.e., 5 years later) another 3 papers on photocatalytic membrane reactors [78–80] were published. Li and Zhao [80] proposed and tested the use of HPC to degrade many different types of dyes present in wastewater coming from the textile dyeing and finishing industry, which were not sufficiently degraded by conventional biological treatment processes. To efficiently separate and reuse the photocatalyst from treated wastewater, the suspended $TiO_2$ powder was separated from slurry by a membrane filter and recycled to the photoreactor continuously. The results demonstrated that the integration of the HPC process with MS permitted us to efficiently degrade non-biodegradable organic substances in the effluent coming from the biological treatment process, completely removing the color from the effluent in a system in which the $TiO_2$ was successfully recovered by a membrane filter and continuously reused in the photoreactor.

After this research, starting from 2000, the attention devoted to the application of photocatalytic membrane reactors by the international scientific community has progressively increased. Simultaneously, the concept of the synergistic coupling between the HPC and the MS processes was becoming more and more widespread. In the previously reported studies, the role of the membrane was simply to efficiently separate and reuse the photocatalyst from treated wastewater. In the early years of the 21st Century, the scientific community was gradually becoming more aware of the dual role of membrane separations: i) confinement of the photocatalyst into the reacting environment, permitting its reuse and helping to achieve the separation and recovery of the photocatalyst from the treated effluent, and ii) control of the contact time between the substrate to be degraded/oxidized/reduced and the photocatalyst. These two roles are the basis of the synergistic effect of coupling HPC and MS in PMRs [81–83].

The increase in interest in the coupling of HPC and MS can be understood by considering the data of the scientific papers documented in the Scopus database under the keywords "photocatalytic membrane reactors", "membrane photoreactors" or "photocatalytic membranes" (Figure 1). The number of published papers per year, individuated with the previously reported keywords, increased from 13 in 2000 to approximately 350 in 2020, with a total of around 2300 papers published over the course of 20 years. During the same time period, the number of citations per year received from the articles concerning PMRs increased quite exponentially from 4 in 2000 to around 13,000 in 2020, with a total of around 70,000 citations in 20 years. Most published papers were under the keyword "photocatalytic membranes".

The first irradiation source widely used in PMR was UV. Only in the last decade have researchers started to modify or develop new photocatalysts to enhance the use of visible light as an irradiation source for photocatalysis coupled with MSs. Figure 2 shows the increasing number of publications on PMRs using visible light as an irradiation source.

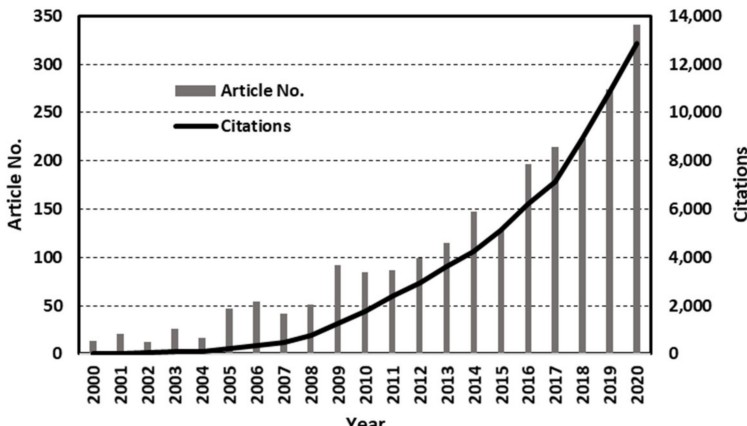

**Figure 1.** Number of articles (Article No.) and related citations regarding PMRs from the year 2000 to 2020 documented in the Scopus database (keywords: "photocatalytic membrane reactor" or "membrane photoreactor" or "photocatalytic membrane").

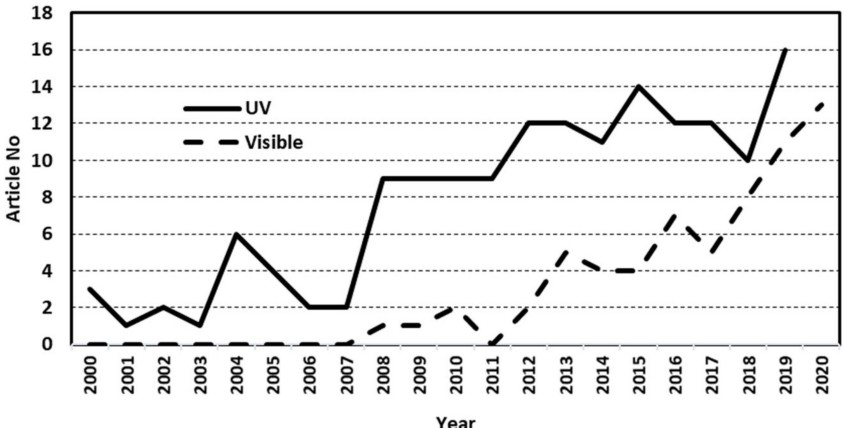

**Figure 2.** Number of articles (Article No.) on the application of PMRs by using UV or visible light related to citations from the year 2000 to 2020 documented in the Scopus database (keywords: "photocatalytic membrane reactor and UV" or "photocatalytic membrane reactor and visible").

A significant amount of knowledge has been gleaned in these years. The progress of knowledge mainly concerns the following aspects: (1) type of membrane, (2) type of coupling photocatalyst and membrane (slurry PMR, PMR with entrapped photocatalysts, pressurized and depressurized systems, etc.), (3) membrane configuration (tubular, flat sheet); (4) membrane processes (pressure driven, membrane distillation, dialysis, pervaporation); (5) activity under solar/visible light; (6) light source type; (7) type of application (degradation and synthesis). These aspects will be developed in the successive sections.

### 3. PMRs in Water and Wastewater Treatment

As previously reported, PMRs have been extensively tested in remediation processes such as water and wastewater treatment, in which organic and inorganic pollutants are totally degraded (mineralized) to innocuous substances.

The interest of the scientific community in PMR application for water and wastewater treatment can be demonstrated by selecting, among the scientific papers documented in the Scopus database previously reported (see Figure 1), the papers containing the keywords "water treatment" or "wastewater treatment" or "degradation" (Figure 3). On this topic, the number of published papers per year increased from 0 in the year 2000 to approximately 120 in 2020, with a total of around 750 papers over the course of 20 years. In the same time interval, the number of citations per year received from those articles increased quite

exponentially from 0 in the year 2000 to around 6000 in 2020, with a total of around 29,000 citations in 20 years.

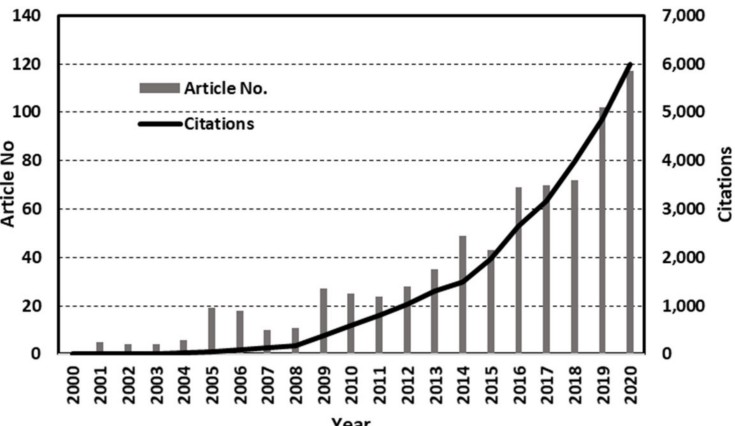

**Figure 3.** Number of articles (Article No.) on water and wastewater treatment and related citations regarding PMRs from the year 2000 to 2020 documented in the Scopus database (keywords: "photo-catalytic membrane reactor" or "membrane photoreactor" or "photocatalytic membrane" and "water treatment" or "wastewater treatment").

### 3.1. PMRs Configuration

One of the first aspects that was considered in integrating HPC with MS in PMRs was the system configuration. As previously reported in the Introduction, two main configurations of PMRs can be distinguished, depending on the "phase" containing the photocatalyst. In 2000, Molinari et al. [84] studied the possibility of using PMRs for water purification. They evaluated the performance of the PMR system for degradation of toxic organic species dissolved in aqueous media. The $TiO_2$ P25 Degussa photocatalyst was immobilized in a flat sheet polymeric membrane and 4-Nitrophenol (4-NP) was used as a model molecule.

Since the photoreactor, and then the membrane, was directly irradiated by the UV lamp, a preliminary investigation into the stability under UV irradiation of some eligible polymeric membranes was carried out. Indeed, in such a system the photocatalytic membrane is constantly exposed to UV irradiation, and the effects of UV irradiation on the structure and long-term stability of polymer membranes deserves to be investigated. Practically, it is important to have membranes resistant to irradiation as well as good photocatalytic performance, thus avoiding undesired membrane photodegradation during the photocatalytic run. Eleven commercial polymeric membranes eligible to be used in the photoreactor were tested [84]. The results showed that membranes made of fluoride + PP (FS 50 PP-Dow), polysulphone + PP (GR 51 PP-TechSep), and polyacrylonitrile (PAN-TechSep) were the best, since they seemed to be quite stable under UV light over 24 h of irradiation. The immobilization of the $TiO_2$ photocatalyst on the membranes was performed by appropriately ultrafiltrating $TiO_2$ suspensions to obtain a deposition of the photocatalyst higher than 2.04 mg of $TiO_2$ per square cm of membrane surface area. The photocatalytic tests demonstrated a 50% degradation of 4-NP after 5 h of irradiation in the presence of air, versus 80% of the suspended photocatalyst reactor. Moreover, an almost complete degradation was observed in the presence of $TiO_2$ suspended in the solution and pure oxygen. The obtained permeate, that is the treated effluent, was clear. The authors demonstrated the possibility of continuous reuse of the photocatalyst and the continuous separation of products from the reaction environments as compared with traditional treatments.

In a successive work, the same group [85] prepared various cellulose triacetate (CTA) and polysulfone (PSF) membranes with entrapped $TiO_2$ by using the phase inversion process. The substantial difference of these membranes with respect to ones prepared

in [84] is that TiO$_2$ photocatalyst is entrapped in the polymeric matrix and not simply deposited on the membrane acting as support. These membranes were tested via their photocatalytic degradation of Congo red, and the obtained photoactivities were compared with the ones obtained by using the same amount of TiO$_2$ in suspension. The results indicated that TiO$_2$ was always more efficient when used in suspension. Then, the authors concluded that the PMR configuration with the photocatalyst in suspension seems to be the better one.

These studies clearly show that one of the first problems with developing PMRs with an immobilized (deposited on/entrapped) photocatalyst is the choice of membrane materials, which can be both organic and inorganic. Indeed, photocatalytic membranes (PMs) can be either made of polymeric [85–88] or inorganic materials [89–97]. Ceramic membranes are often utilized in PMRs with an entrapped photocatalyst because of their higher resistance to UV irradiation and oxidative species [89,98].

In this type of study, Dzinun et al. [99] investigated the stability of a TiO$_2$/PVDF dual layer hollow fiber membrane against photocatalytic reactions for photocatalytic-membrane process. It was noticed that the tensile strength of used TiO$_2$/PVDF membranes decreased moderately after 30 days of UV irradiation, which results in negative impacts on the membrane's overall stability. This work furtherly demonstrates that is of particularly importance to evaluate the sustainability of polymeric membrane, which has considered as the heart of a photocatalytic membrane reactor for a wastewater treatment process.

Alias et al. [100] prepared and tested porous photocatalytic ceramic membranes for humic acid photodegradation. They prepared some ceramic photocatalytic membranes by using the dip-coating method. First, they prepared the supports, i.e., ceramic membranes, by phase inversion of a casting solution containing poly ether sulfone (PES) as the binder, N-methyl-2-pyrrolidone (NMP) as the solvent, and kaolin powder as the membrane forming component. The so-prepared membrane supports were dip-coated with a suspension containing TiO$_2$ nanoparticles and poly(ethylene glycol) (PEG). The so-obtained photocatalytic membrane was successively dried and sintered. These photocatalytic membranes exhibited good antifouling and self-cleaning performance and a humic acid rejection of 98.56% subsequent to UV light exposure.

A PMR with a photocatalytic membrane prepared by coating the TiO$_2$ catalyst on the surface of a porous ceramic tube, was studied by Wang et al. [101] in the photodegradation of acid red 4 (AR4) dye from aqueous media. In this system, the permeation of solutes through the membrane and the photocatalytic reaction occur simultaneously. All photocatalytic experiments were conducted in one pass dead-end system. Some tests were also carried out to compare the photocatalytic degradation of AR4 in a dead-end and a cross-flow system. The main results of this study were: (i) the photocatalytic degradation for the dead-end system was three–five times higher than the cross-flow system depending on flow rates; (ii) the decomposition ratio decreased by increasing the flow rate; (iii) the decomposition ratio increased with increasing catalyst loading and light intensity, but remained constant at a catalyst loading higher than a limit value.

As previously reported, the problem related to the photostability of polymeric membranes in a TiO$_2$ based photocatalytic process is important in all the integrative-type PMRs, where the HPC and the MS take place in the same apparatus and thus the membrane is directly irradiated. PMRs with entrapped photocatalysts are intrinsically integrative-type, but slurry PMRs can also be integrative. On this basis, China et al. [102] made a study on the selection of polymeric membranes to be used in an integrated slurry type PMR. Ten membranes were evaluated under ultraviolet (UV) and TiO$_2$ photocatalysis conditions. Membrane stability was measured in terms of changes in pure water flux (PWF), release of total organic carbon (TOC), and scanning electron microscope (SEM) morphology analyses. The results showed that polytetrafluoroethylene (PTFE), hydrophobic polyvinylidene fluoride (PVDF) and polyacrylonitrile (PAN) membranes showed the greatest stability.

The results obtained by Molinari et al. [84,85] confirm what was found by Enzweiler et al. [77], showing that slurry PMRs make it possible to obtain higher efficiency with respect

to the immobilized system. This trend can be ascribed to the larger active surface area in the case of slurry PMRs, resulting in a higher surface area between the photocatalyst and the substrates to be degraded (the pollutants). Despite this limitation, in the immobilized PMRs, the recovery and reuse of the photocatalyst is easier. Furthermore, systems with the photocatalyst entrapped in/on the membrane are usually characterized by better performance with respect to conventional membranes in terms of decreased membrane fouling and enhanced permeate quality.

This trend was confirmed in numerous works. Geissen et al. in 2001 [103] made a comparison between a photocatalytic membrane reactor with the same suspended and fixed photocatalyst, showing that whereas for the fixed system no separation step is necessary and a simple construction can be used, suspended systems offer a three times higher reaction rate for the same photocatalyst concentration, but they are also characterized by higher investment costs. Molinari et al. [104] compared the experimental results obtained by using various configurations of photocatalytic membrane reactors (PMRs) in water purification using 4-nitrophenol as a model pollutant. The results demonstrated that the split-type configuration appeared to be the most interesting for industrial applications. Indeed, in such a system the HPC and MS process take place in different apparatus but if they are adequately coupled, high irradiation efficiency, high membrane permeate flowrate and selectivity can be obtained by optimal sizing of the separate "photoreactor" and the "membrane cell" by taking advantage of all the best research results for each of these two units. The same group compared different system configurations in the photocatalytic degradation of two commercial azo-dyes, i.e., Congo red and Patent blue, in aqueous solution by using $TiO_2$ P25 as the photocatalyst [105]. A comparison between suspended and entrapped $TiO_2$ was also done in the two different experimental set-ups. The first system is a split-type slurry PMR with the lamp immersed in the photoreactor, while the second one is an integrative-type PMR with entrapped photocatalyst and external lamp. These systems are different in terms of configuration, the mode of using the photocatalyst and the kind of radiation source and its position with respect to the reacting mixture.

The results confirmed that the PMR configuration containing the suspended photocatalyst was significantly more efficient than the configuration where the photocatalyst was entrapped in the membrane. Moreover, the immersed UV lamp gave a photodegradation rate of Congo red in the batch photoreactor without membrane around 50 times higher than that found with the external lamp. By comparing the performance of the batch system without the membrane (i.e., the photoreactor) with that of the PMR with the membrane, a lower degradation rate was observed in the PMR, since a significant volume of the solution containing the pollutant and the photocatalyst was not continuously irradiated. To overcome this limitation, a minimization of the volume of the recycling loop with respect to the irradiated volume in the photocatalytic membrane reactor was required. Moreover, it was demonstrated that by operating with the suspended photocatalyst it was possible to successfully treat highly concentrated solutions (500 mg $L^{-1}$) of both dyes by means of a continuous process obtaining good values of permeate fluxes, which is interesting in view of industrial applications. Thus, the PMRs permit us to continuously operate in a system in which pollutant photodegradation, photocatalyst recovery and reuse and separation of substances from the treated solution simultaneously occur.

Many other studies have demonstrated that PMRs with a suspended photocatalyst permit us to obtain higher efficiency in comparison with PMRs with immobilized photocatalyst [44,106]. Thus, slurry PMRs have been widely experimented with in the literature [107–109]. However, the performance of this kind of PMR is limited by two factors. The first one is light scattering by the suspended photocatalyst nanoparticles (NPs), and the second one is membrane fouling, caused by the deposition of the photocatalyst NPs on the membrane surface, resulting in a permeate flux decline [110–113].

A first approach to limit membrane fouling consists in properly selecting hydraulic conditions in the membrane module. Based on this idea, Fu et al. [114] reported the photodegradation of fulvic acid (FA) by using home-made nanostructured $TiO_2$/silica

gel photocatalyst particles. The photocatalytic tests were performed in the submerged PMR (SPMR) schematized in Figure 4. In this system, the photocatalytic degradation of fulvic acid is coupled with a membrane filtration operated by a depressurized submerged membrane module. Air is bubbled under the membrane module, having a dual role: limiting the accumulation of the photocatalyst on the membrane surface and feeding the oxygen needed for the photocatalytic process.

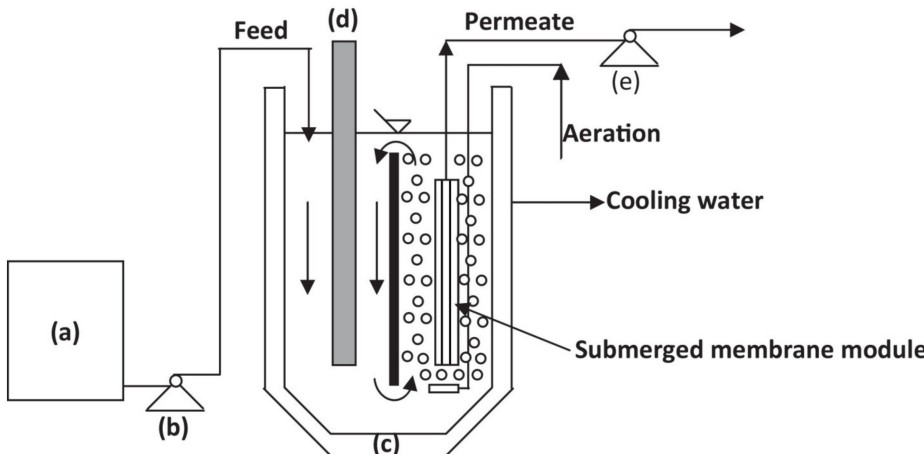

**Figure 4.** Schematic diagram of the SMPR system: (**a**) feed tank; (**b**) feed pump; (**c**) thermostatic jacket photoreactor; (**d**) UV lamp; (**e**) suction pump (elaborated from [114]).

The results demonstrated that the proposed system effectively reduced photocatalyst accumulation on the membrane surface. Photocatalyst concentration and air flow represent the main operational parameters characteristic of this system affecting system performance. It was found that 0.5 g L$^{-1}$ photocatalyst concentration and 0.06 m$^3$ h$^{-1}$ air flow were the optimal conditions for the removal of FA. Moreover, the FA degradation rate was higher in acidic conditions than in basic conditions. Bare TiO$_2$ P25 and nano-structured TiO$_2$ were compared in terms of induced membrane fouling. According to the experimental results, the permeating flux increased by using nano-structured TiO$_2$ photocatalyst, showing that its use permitted us to reduce membrane fouling. Based on these results, it was concluded that the use of SPMRs with air bubbling and adequate membrane back-flushing is a promising approach to limiting membrane fouling.

A similar approach was tested by Zheng et al. [48] in the removal of viruses from aqueous media. The experimental tests were performed by using the experimental set-up schematized in Figure 5 under constant flux mode. The SPMR was assembled as follows: (i) a photoreactor consisting of a vessel with a total volume of 12.75 L in which a 4 W UV-C lamp emitting at 254 nm wavelength was immersed; (ii) a membrane module, containing a flat-sheet PVDF membrane (average pore size 0.15 μm, membrane area 0.03 m$^2$), immersed into the photoreactor; (iii) an aeration system, which fed air bubbles at an aeration rate of 10 L min$^{-1}$; (iv) a programmable logic controller (PLC) for controlling the system during operations; (v) a feed tank for continuous operation. The temperature of the aqueous solution, recirculated by a pump, was maintained in the range 20–25 °C.

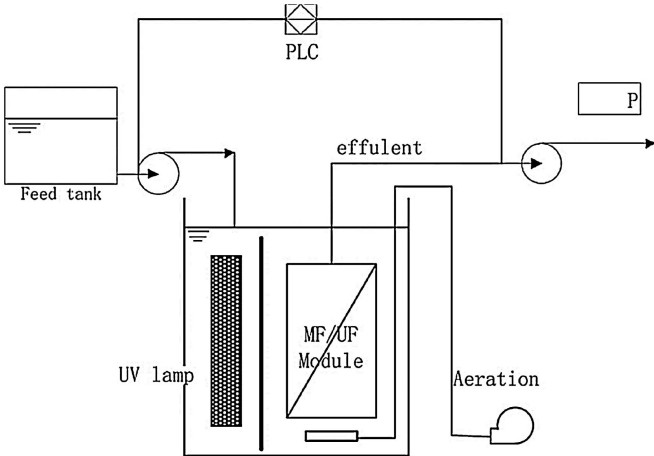

**Figure 5.** Schematization of the experimental SPMR used by Zheng et al. Reprinted with permission from ref. [48]. Copyright © 2021 Elsevier B.V.

The virus bacteriophage f2, which is characterized by a dimension like that of human enteric viruses, was used as the model virus. Nano-TiO$_2$ P25 was used as the photocatalyst. The main operating conditions influencing system performance, i.e., filtration flux and permeation mode (continuous or intermittent), were tested. It was found that the optimal operating condition was the intermittent suction mode with a filtration flux of 40 L m$^{-2}$ h$^{-1}$, which gave good residence time in the photoreactor, satisfactory photodegradation, and a reasonable control of membrane fouling. Above this "critical" value of the filtration flux, irreversible fouling was observed. An average virus removal equal to 99.999% was achieved by operating in continuous mode for 24 h. This result demonstrated that the proposed SPMR made it possible to obtain the inactivation of the virus thanks to the action of OH radicals and to the membrane, which permitted the maintaining into the reacting environment of both photocatalyst and virus.

A conceptually similar system (Figure 6) was also studied by Kertèsz et al. [115]. This system was tested in the photocatalytic degradation of acid red 1 (AR1), by using TiO$_2$ in suspension.

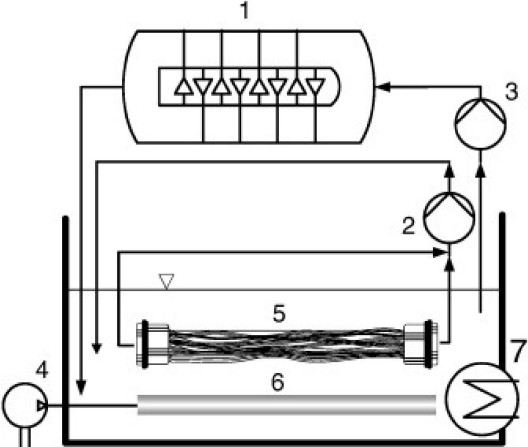

**Figure 6.** Schematization of the SPMR used by Kertèsz et al. [115]: (1) photoreactor, (2) permeate pump, (3) circulation pump, (4) air compressor, (5) hollow fiber membrane (HFM) module, (6) air diffuser, (7) temperature control system. Reprinted with permission from ref. [115]. Copyright © 2021 Elsevier B.V.

The results demonstrated that wastewater containing AR1 was successfully decolorized and simultaneously the complete photocatalyst recovery was obtained. The effectiveness of air bubbling in controlling the rapid flux decline, caused also by the dead-end suction filtration mode, was confirmed. A critical permeate flux equal to 40 L m$^{-2}$ h$^{-1}$ was found. Below this permeate flux membrane fouling was reversible, and membrane back-flushing with the permeate permitted us to easily repristinate membrane performance, while above this value irreversible fouling was detected. This trend was also reported by Zheng et al. [48]. The authors also demonstrated that membrane fouling can be controlled mainly by optimizing the frequency, duration, and intensity of membrane back-flushing.

A depressurized SPMR with suspended and immobilized N–TiO$_2$ photocatalyst was tested by Nguyen et al. [41] for diclofenac (DCF) removal from wastewater under visible irradiation. It was shown that initial photocatalyst concentration significantly affected the DCF removal efficiency. The best removal efficiency was achieved at photocatalyst concentration of 1.5 g L$^{-1}$. A comparison between the SPMRs with suspended and immobilized N–TiO$_2$ showed that the SPMR with the suspended catalyst showed better DCF removal efficiency because the N–TiO$_2$ suspended particles increased DCF removal. The author emphasized that the SPMRs with suspended and immobilized N–TiO$_2$ have both advantages and disadvantages. An advantage of the SPMR with suspended photocatalyst is that the reaction rate can be enhanced by increasing the N–TiO$_2$ dosage. As the downside, the configuration with suspended photocatalyst showed higher membrane fouling than the SPMR with immobilized photocatalyst, demonstrated by a faster decrease in the permeate flux. Some tests were carried out in a continuous system in which an RO membrane was combined with the SPMR (Figure 7). This combination resulted in good effluent quality, but the DCF and TOC concentrations in the photoreactor improved because DCF and its degradation by-products were recirculated to the photoreactor by RO rejection.

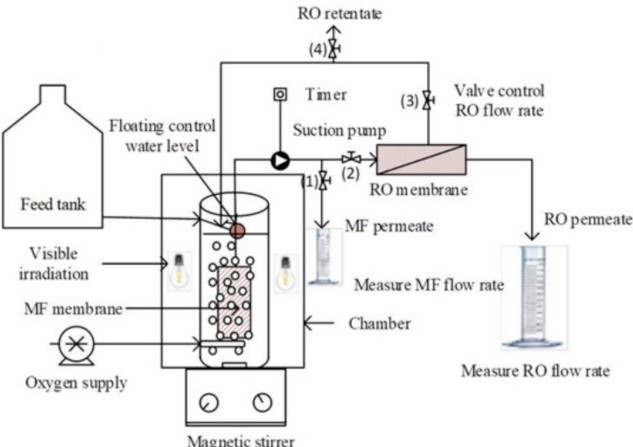

**Figure 7.** Schematization of the continuous SPMR+RO system used by Nguyen et al. Reprinted with permission from ref. [41]. Copyright © 2021 Institution of Chemical Engineers. Published by Elsevier B.V.

The treatment of p-nitrophenol (PNP) polluted wastewater by using a submerged photocatalytic membrane reactor with a visible-light responsive Fe(III)-ZnS/g-C$_3$N$_4$ photo-Fenton catalyst was very recently proposed and studied by Wang et al. [116] in the system schematized in Figure 8. The key operating parameters were successively optimized to obtain a 91.6% PNP removal by operating the SMPR under simulated solar light irradiation with 10 mg L$^{-1}$ PNP concentration in the feed, initial pH 5, catalyst dosage 1.0 g L$^{-1}$, H$_2$O$_2$ concentration 170 mg L$^{-1}$, aeration rate 0.50 m$^3$ h$^{-1}$ after 4 h of irradiation. The photocatalyst was completely rejected by the MF membrane, thus realizing the rapid separation and recycling of the suspended photocatalyst. The overall toxicity of the treated solution decreased after the visible-light-driven photo-Fenton reaction. Thus, it was confirmed that the combination of HPC and MS permits us to obtain a system realizing a

high treatment efficiency of refractory wastewater pollutants while solving the problem of the separation and recycling of the powder photocatalyst.

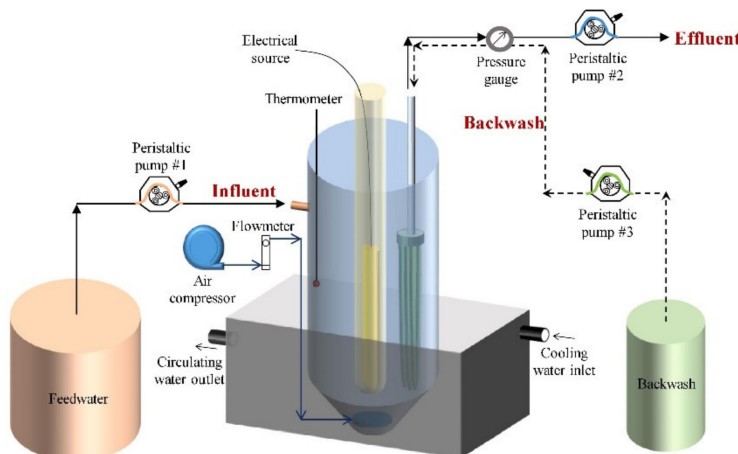

**Figure 8.** Schematic diagram of the combined photocatalysis-membrane filtration process used by Wang et al. Reprinted with permission from ref. [116]. Copyright © 2021 Elsevier B.V.

The described results demonstrated that performance of PMRs with suspended photocatalysts can be enhanced by using submerged membrane modules operated in de-pressurized mode, coupled with air bubbling and membrane back-flushing. In this way, the bigger limits of slurry PMR, i.e., membrane fouling, is controlled. Thus, this type of hybrid system has been especially widely investigated in recent years [40,117–121].

Recently, Jafri et al. [122] studied the use of hollow titanium dioxide nanofibers (HTNF) for photocatalytic degradation of organic pollutants. The photocatalytic performance of HTNF were studied by evaluating the degradation of Bisphenol A (BPA) under UV light irradiation. In the photodegradation process, the effect of several parameters such as the initial concentration of BPA, pH of the solution, and photocatalyst dosage were investigated. The optimum photocatalyst dosage, pH and initial BPA concentration were 0.75 g L$^{-1}$, pH 4.1 and 10 ppm BPA, respectively. Under these conditions, the photocatalytic degradation of BPA was found to be 97.3% by using HTNF, which was 12.6% higher than those of Degussa P25 TiO$_2$. The degradation of BPA followed the pseudo-first-order kinetic model.

Very recently, Zhang et al. [123] reported a synergistic system of integrated photocatalysis-adsorption-membrane separation in a rotating reactor (Figure 9). This system contains a composite membrane consisting of graphene oxide (GO) acting as the separation membrane, activated carbon (AC) as the adsorbent, and Ag@BiOBr as the photocatalyst, respectively. In this system, the GO membrane layer could reject the organic molecules with the assistance of AC layer with efficient adsorption capacity, and Ag@BiOBr at outer layer could photodegrade the organics under visible light irradiation. The authors reported that the rejection rate of RhB in the case of Ag@BiOBr/AC/GO membrane always maintained up to about 100%, compared to the gradually decreased rejection rate on AC/GO. It indicated that the Ag@BiOBr photocatalyst loaded on the membrane surface could degrade the adsorbed pollutants and thus decrease the membrane fouling.

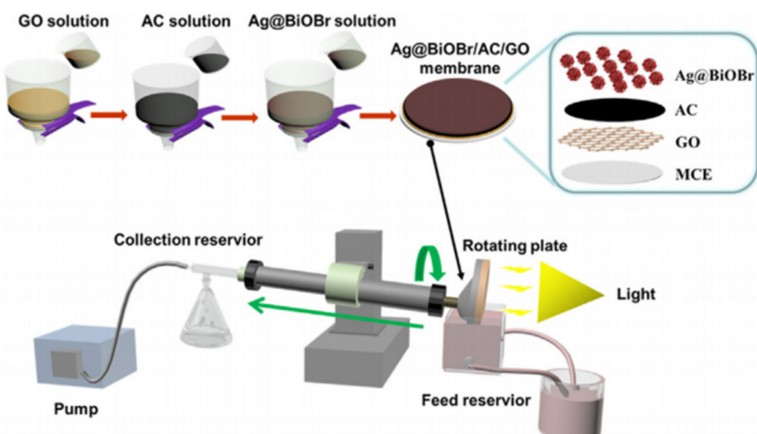

**Figure 9.** Illustration of rotating reactor and preparation of Ag@BiOBr/AC/GO membrane by Zhan et al. Reprinted with permission from ref. [123]. Copyright © 2021 Elsevier Ltd.

Singh et al. [124] reported the possibility of coupling advanced oxidation technology and membrane processes during chloramphenicol (CAP) filtration by using a low-pressure cross-flow lab-scale photocatalytic membrane reactor (PMR). The photocatalyst used for chloramphenicol (CAP) degradation was Titanium dioxide doped hydroxyapatite ($TiO_2$-HAP). To enhance the photocatalytic activity and antifouling propensity, different amounts of $TiO_2$-HAP photocatalyst were incorporated on polysulfone (PSf) membranes. The authors reported the highest degradation of 61.59% for the PSf/4 wt% $TiO_2$-HAP nanocomposite membrane.

In Table 1, some main results concerning the evolution of water treatment using PMRs in various configurations coupling the photocatalyst with the membrane are reported. The results clearly demonstrate that the use of PMR permits us to continuously operate in a system in which efficient water treatment and photocatalyst recovery simultaneously occur. Moreover, an important aspect to be considered is the choice of PMR configuration for water treatment. Slurry PMR permits us to obtain better performance with respect to PMR using an entrapped photocatalyst. Moreover, split type PMR, in which the photocatalytic and the membrane separation steps take place in different vessels, appears to be the most promising in view of large-scale applications: indeed, for such systems the photocatalytic and the membrane separation steps can be separately studied and optimized, which is an important advantage from an engineering point of view. The bigger limits of slurry PMR, which are membrane fouling and light scattering by photocatalyst particles, can be limited by using SPMRs with air bubbling and periodic membrane back flushing, resulting in a system with good potential for large scale applications. During the implementation of these systems, it must be taken into account that photocatalyst concentration and air flow significantly affect system performance. Moreover, considering that a "critical value" of the filtration flux exists, above which irreversible fouling takes place, the permeation flux must be adequately chosen.



**Table 1.** Summary of various pressure-driven PMR configurations in water treatment and their main results.

| PMR Configuration | Support | Photocatalyst | Pollutant | Main Results | Ref. Year |
|---|---|---|---|---|---|
| Photocatalyst immobilized on the membrane | 11 commercial polymeric membranes | $TiO_2$ P25 | 4-nitrophenol | 50% photodegradation after 5 h with immobilized photocatalyst 80% photodegradation with suspended photocatalyst | [84] 2000 |
| Three integrative-type PMRs vs. a split type PMR | 11 commercial polymeric membranes | $TiO_2$ P25 | 4-nitrophenol | Split-type configuration appeared to be the most promising for industrial applications: PMR optimization can be obtained by sizing separately the "photoreactor" and the "membrane cell". Limits: membrane fouling and light scattering by photocatalyst particle | [104] 2002 |
| Suspended vs. entrapped $TiO_2$ | NTR7410 membrane vs. home prepared photocatalytic membrane | $TiO_2$ P25 | Congo red Patent Blue | Slurry PMR was significantly more efficient than the PMR with entrapped photocatalyst. Solutions with high concentration of dyes can be treated by a continuous process obtaining good permeate fluxes and quality. Limit: membrane fouling | [103] 2004 |
| Photocatalyst entrapped in the membrane | Cellulose triacetate (CTA) and polysulfone (PSF) membranes | $TiO_2$ P25 | Congo red | $TiO_2$ was always more efficient when used in suspension | [85] 2005 |
| Slurry integrative-type PMR | 10 polymeric membranes | $TiO_2$ | - | polytetrafluoroethylene (PTFE), hydrophobic polyvinylidene fluoride (PVDF) and polyacrylonitrile (PAN) membranes showed the greatest stability. Limits: membrane fouling and light scattering by photocatalyst particle. | [102] 2006 |
| Submerged PMR air bubbling | Submerged hollow fiber module | nanostructured $TiO_2$/silica gel | Fulvic acid | Effective reduction in membrane fouling. Photocatalyst concentration and air flow significantly affect system performance | [114] 2006 |
| Photocatalyst coated on the membrane | Porous ceramic tube | $TiO_2$ | Acid Red 4 | Photodegradation obtained with the dead-end system was three/five times higher than cross-flow system. Increasing photodegradation with increasing catalyst loading and light intensity, to a catalyst loading limiting value. | [101] 2008 |
| Submerged PMR air bubbling and membrane back-flushing | Submerged hollow fiber membrane (HFM) module | $TiO_2$ | Acid Red 1 (AR1) | Simultaneous AR1 degradation and complete photocatalyst recovery. Air bubbling was effective in controlling membrane fouling. Critical permeate flux 40 L m$^{-2}$ h$^{-1}$. Flux < 40 L m$^{-2}$ h$^{-1}$ gave reversible fouling, easily removed by membrane back-flushing with the permeate, Flux > 40 L m$^{-2}$ h$^{-1}$ gave irreversible fouling. The control of membrane fouling depends mainly by membrane back-flushing parameters, i.e., frequency, duration and intensity. | [115] 2014 |

Table 1. *Cont.*

| PMR Configuration | Support | Photocatalyst | Pollutant | Main Results | Ref. Year |
|---|---|---|---|---|---|
| Submerged PMR air bubbling and membrane back-flushing | Flat-sheet PVDF membrane | TiO$_2$ P25 | Virus bacteriophage f2 | Filtration flux and permeation mode (continuous or intermittent), significantly affect system performance. Best operating conditions: intermittent suction mode, filtration flux of 40 L m$^{-2}$ h$^{-1}$, 99.99% virus inactivation, good control of membrane fouling. Above the "critical" value of the filtration flux, irreversible fouling was observed. | [48] 2015 |
| Photocatalyst entrapped in the membrane | Polyvinylidene difluoride (PVDF) | TiO$_2$ | - | Limited membrane stability: the tensile strength of the TiO$_2$/PVDF membranes decreased after 30 days of UV irradiation | [99] 2017 |
| Photocatalyst coated on the membrane | Kaolin powder | TiO$_2$ nanoparticles | Humic acids | 98.6% photodegradation good antifouling self-cleaning performance good membrane photostability | [100] 2018 |
| Submerged PMR with suspended and immobilized photocatalyst and air bubbling | MF ceramic membrane | N–TiO$_2$ | Diclofenac | SPMR with suspended catalyst showed better DCF removal. SPMRs with suspended and immobilized N–TiO$_2$ have both advantages and disadvantages. Advantage of slurry-SPMR: the reaction rate can be enhanced by increasing the photocatalyst. Disadvantage of slurry-SPMR: higher membrane fouling. | [41] 2020 |
| SPMR | Hollow fiber microfiltration (MF) membrane module | Fe(III)-ZnS/g-C$_3$N$_4$ photo-Fenton catalyst | p-nitrophenol (PNP) | 91.6% PNP under simulated solar light irradiation, 10 mg L$^{-1}$ PNP concentration in the feed, initial pH 5, catalyst dosage 1.0 g L$^{-1}$, H$_2$O$_2$ concentration 170 mg L$^{-1}$, aeration rate 0.50 m$^3$ h$^{-1}$, 4 h of irradiation. The photocatalyst was completely rejected by the MF membrane. | [116] 2021 |
| Hollow titanium dioxide nanofibers (HTNF) | Polyacrylonitrile (PAN) nanofibers | TiO$_2$ | Bisphenol A (BPA) | photocatalytic degradation of BPA 97.3% | [122] 2021 |
| Integrated photocatalysis-adsorption-membrane separation in a rotating reactor | GO | Ag@BiOBr | RhB | The rejection rate of RhB in the case of Ag@BiOBr/AC/GO membrane was always maintained up to about 100% | [123] 2021 |
| Low-pressure cross-flow lab-scale photocatalytic membrane reactor (PMR) | Polysulfone (PSf) membranes | TiO$_2$-HAP | Chloramphenicol (CAP) | Degradation of 61.59% for the PSf/4 wt% TiO$_2$-HAP nanocomposite membrane. | [124] 2021 |

### 3.2. Combination of HPC with Other Membrane Processes

The applications of PMRs described in the previous section are referred to systems in which the HPC process is coupled with the so-called pressure driven membrane processes, i.e., microfiltration (MF), ultrafiltration (UF), nanofiltration (NF), and reverse osmosis (RO). These processes differ in the size of the solutes retained by the membrane, but they have one common feature: the driving force promoting the flux across the membrane and then the separation, is a pressure difference. As demonstrated, this coupling usually gave better results in the slurry-type configuration, because the larger active surface area guarantees a good contact between the photocatalyst and the substrate to be degraded. Moreover, the slurry-type configuration has the advantage that the reaction rate can be enhanced by increasing the photocatalyst concentration. As a downside, however, by increasing photocatalyst concentration, light scattering and membrane fouling increases, resulting in a decreased performance of both the coupled processes: (i) HPC because of light scattering, and (ii) MS because of membrane fouling. These effects of membrane fouling, which are intrinsic limitations of the traditional slurry-type PMRs, can be prevented by coupling photocatalysis with MSs that are based on driving forces different from the pressure and different transport mechanisms.

Moving in this direction, a first possible approach consists in coupling HPC and membrane distillation (MD). This one is a separation process based on vapour–liquid equilibrium in which: (i) the non-volatile components (e.g., ions, macromolecules, etc.) are retained on the feed side; (ii) the volatile components pass through a porous hydrophobic membrane and then they condense in a cold distillate (usually distilled water).

In a series of studies, Mozia et al. [58,125–131] proposed and tested the use of a PMR obtained by coupling HPC with direct contact membrane distillation (DCMD). In [58], the effectiveness of the proposed PMR was evaluated with the removal of three different azo dyes (Acid Red18 (AR18), Acid Yellow 36 (AY36) and Direct Green 99 (DG99)) contained in aqueous solution. $TiO_2$-P25 was the photocatalyst and the membrane module was equipped with 9 polypropylene capillary membranes. The membranes guaranteed an effective area of 0.014 $m^2$ and permitted us to maintain the photocatalyst and the pollutants in the feed compartment. By operating with this PMR, it was demonstrated that the presence of $TiO_2$ did not affect the permeate flux. In particular, a 0.34 $m^3\,m^{-2}\,d^{-1}$ permeate flux, equal to that one obtained by using ultrapure water as feed, was obtained regardless of $TiO_2$ concentration. Moreover, the MD process was very effective in rejecting the photocatalyst particles, the dye and other non-volatile compounds (organic molecules and inorganic ions), so that the turbidity of distillate was similar to that measured for ultrapure water, regardless of the $TiO_2$ loading used. Some volatile organic compounds permeated across the membrane, as demonstrated by the total organic carbon (TOC) values measured into the permeate/distillate. However, the distillate was practically pure water since TOC was in the range 0.4–1.0 mg $L^{-1}$. Based on these results, it can be affirmed that the MD membrane acts as an effective barrier for the photocatalyst nanoparticles and also for the non-volatile compounds present in the feed. Therefore, the proposed PMR, obtained by coupling HPC and DCMD, could be another method for the removal of organic compounds from water [58].

The same concept, i.e., a PMR coupling DCMD and HPC induced by UVC radiation, was applied for the removal of some non-steroidal anti-inflammatory drugs (diclofenac, ibuprofen (IBU) and naproxen (NAP) sodium salts) from different aqueous matrices [132] (ultrapure water, tap water, primary and secondary effluents of municipal wastewater treatment plant). It was observed that the efficiency of drug removal depends on the feed matrix. The photodegradation efficiency followed the order: ultrapure water > tap water (TW) > secondary effluent (SE) > primary effluent (PE). In the worst case, i.e., in the case of the primary effluent, by operating with a $TiO_2$ P25 photocatalyst loading of 0.5 g $L^{-1}$ a complete DCF removal was obtained, while IBU concentration decreased by 73% and NAP by 90%. In the case of SE and operating with a $TiO_2$ P25 photocatalyst loading of 1.5 g $L^{-1}$, higher values were obtained (DCF 100%, IBU 93% and NAP 94%). Despite these good

degradation values, less encouraging mineralization (i.e., complete degradation to $CO_2$) was obtained (14% and 23% for PE and SE, respectively). No drugs were detected into distillate, the removal of DOC was higher than 99% for both PE and SE, and no permeate flux decline was observed for TW and SE. However, during PE treatment a 50–60% flux decline was observed. These results demonstrated that the hybrid HPC-MD system can be an effective technology for the removal of pharmaceuticals from SE and PE. However, because of membrane fouling, the PE should be pre-treated before feeding the PMR.

A novel PMR was designed by Hou et al. [133] coupling Ag/BiOBr visible-light photocatalysis with membrane distillation. In this system (Figure 10), with the aim to avoid light-shielding effect from colored solution containing dye pollutant, Ag/BiOBr films were coated on glass substrates in a thin rectangular wastewater tank.

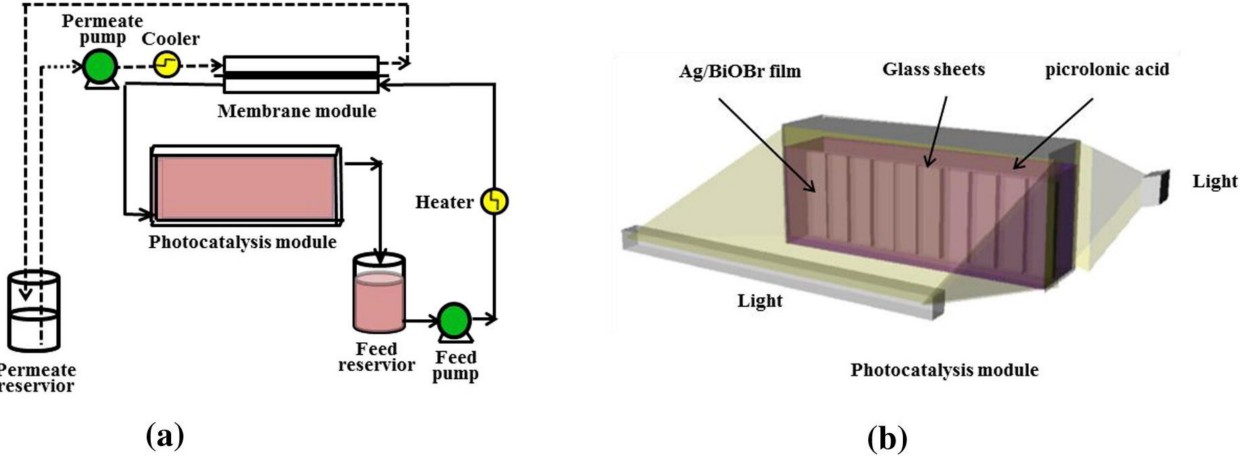

**(a)**                                                                                           **(b)**

**Figure 10.** Schematization of the DCMD/PMR (**a**) and the inside photocatalysis module (**b**) proposed by Hou et al. Reprinted with permission from ref. [133]. Copyright © 2021 Elsevier B.V.

Ag-modification promoted the photocatalytic process under visible light and facilitated the separation of photo-induced electrons from holes thus inhibiting electron/hole recombination. This PMR was used for treating wastewater containing picrolonic acid (PCA). The results demonstrated that the Ag/BiOBr photocatalyst mineralized PCA into $CO_2$ and inorganic nitrogen species (such as $NO_2^-$, $NO_3^-$, and $NH_4^+$) under visible light irradiation. Simultaneously, MD permitted us to produce high-quality water as the distillate. The used polytetrafluoroethylene (PTFE) membrane stopped the passage of PCA and nitrogen species into the distillate due to its hydrophobic property and the non-volatile nature of both PCA and nitrogen species.

Summarizing, the reported results on the application of PMRs obtained by coupling HPC with MD demonstrated that this system is a promising technique for treating wastewaters containing organic pollutants such as azo dyes or pharmaceuticals. Indeed, MD efficiently rejects both photocatalyst particles and organic contaminants contained in the feed solutions and permits us to limit membrane fouling. The relatively low distillate flux assured the needed residence time of substrate to be degraded in the photoreactor thus resulting in high photodegradation efficiency. The high energetic demand of MD represents a significant cost to be included when comparing the PMR/DCMD process with PMR/UF or PMR/NF.

As previously reported, pressure-driven membrane techniques can lead to membrane fouling while the combination of HPC and MD avoids this problem, but it needs energy to reach evaporation phenomena. Due to these limitations, the combination of dialysis with HPC was proposed. In particular, Azrague et al. [60] tested a PMR obtained by the synergistic coupling of HPC and dialysis in the mineralization of 2,4-dihydroxybenzoic acid (2,4-DHBA) as model organic pollutant contained in turbid waters. The proposed PMR has the advantage of working at ambient temperature with the membrane acting as a

barrier for the photocatalyst particles and allowing to extract the organic compounds from the turbid water thanks to the concentration difference between the two compartments. No transmembrane pressure TMP between the feed and strip compartments exists. The absence of TMP avoids fouling problems. In this system the membrane permits us to maintain the TiO$_2$ photocatalyst in the photocatalytic compartment, thus obtaining a clean treated water, and it also permits the maintaining of bentonite away from the photoreactor, thus avoiding light scattering. These advantages, combined with the complete removal of the organic pollutant, demonstrated the effectiveness of the proposed PMR combining HPC and dialysis. Despite these potentialities, this system was not considered elsewhere.

Another possible combination between HPC and membrane processes was tested by Camera-Roda and Santarelli [134]. The integration between HPC and pervaporation (PV) was tested in the detoxification of waters polluted with low concentrations of recalcitrant organic compounds (e.g., 4-chlorophenol (4-CP)). The results showed that the degradation rate of 4-CP was highly improved by integrating HPC and PV with respect to the following alternatives: HPC alone, PV alone, HPC and PV in series, without coupling. Therefore, it was demonstrated that the synergy between HPC and PV gives process intensification. Practically, HPC is positively influenced by PV because the membrane continuously removes some intermediate products that could negatively affect the kinetics of the photocatalytic reaction. Simultaneously, the HPC step positively influences the PV step since HPC converts the feebly permeable 4-CP into organic intermediates that can be removed by PV at a high rate, i.e., hydroquinone and especially benzoquinone. Despite these encouraging results, this PMR possess the following significant drawbacks: (i) around 50% 4-CP degradation in the retentate after 30 h; (ii) the photodegradation intermediates, i.e., hydroquinone and especially benzoquinone, are removed from the reacting environment to the permeate at a high rate by PV, resulting in limited mineralization because of the insufficient residence time in the photoreactor; (iii) the permeate, containing these by-products, requires opportune treatment.

The second limitation, i.e., the limited residence time of some intermediates in the reacting environment, represents an important limit considering the environmental applications, where it is fundamental to obtain complete oxidation of the pollutant to be removed. On the other hand, considering HPC application in organic synthesis, this aspect can become a significant advantage: indeed, when the objective of the photocatalytic process is to partially oxidize or reduce a substrate, the limited residence time in the photoreactor could result in good process selectivity.

By comparing the results obtained by coupling HPC and pressure driven membrane processes (Table 1) with the overall results described in the present section (Table 2), it can be concluded that, at present, SPMRs with air bubbling and membrane back flushing appears to have greater potential for large scale application. Considering that the most studied alternative approach to pressurized driven membrane processes, MD, despite the membrane being very effective at rejecting the pollutants and the photocatalyst particles, permits the limiting of the membrane fouling to obtain a high quality permeate, the high energetic consumption makes this alternative less than ideal for water treatment.

**Table 2.** Summary of various non-pressure driven PMR configurations in water treatment and their main results.

| PMR Configuration | Support | Photocatalyst | Pollutant | Main Results | Ref. Year |
|---|---|---|---|---|---|
| HPC + DCMD | membrane module 9 polypropylene capillary membranes | $TiO_2$-P25 | Acid Red18 (AR18) Acid Yellow 36 (AY36) Direct Green 99 (DG99) | The presence of $TiO_2$ and dye did not affect the permeate flux, regardless of $TiO_2$ and dye concentrations. The MD step was very effective in rejecting the photocatalyst particles and the dye and other non-volatile compounds: so, the turbidity of distillate was similar to that of ultrapure water, regardless of the $TiO_2$ concentrations. The high energetic consumption of MD must be considered. | [58] 2007 |
| HPC + dialysis | hollow fibers module (polyacrylonitrile or polysulfone) plate and frame module (cellophane) | $TiO_2$-P25 | 2,4-dihydroxybenzoic acid (2,4-DHBA) | Advantage 1: operation at ambient temperature. Advantage 2: no transmembrane pressure TMP → no membrane fouling. The membrane allows to maintain the $TiO_2$ photocatalyst in the photocatalytic compartment and allows to extract the organic compounds from the turbid water. Despite these potentialities, this system was not considered elsewhere. | [60] 2007 |
| HPC + PV | GFT Sulzer Chemtech MEM 1070 | $TiO_2$-P25 | 4-chlorophenol (4-CP) | PV positively influences HPC, and concurrently the PV takes advantage from the HPC. Drawback 1: around 50% 4-CP degradation. Drawback 2: the photodegradation intermediates are removed from the reacting environment to the permeate at a high PV rate, resulting in insufficient mineralization because of the limited residence time into the photoreactor. Drawback 3: the permeate solution, containing these by-products, need to be opportunely treated. | [134] 2007 |
| HPC + DCMD | membrane module 9 polypropylene capillary membranes | $TiO_2$-P25 | diclofenac, ibuprofen, and naproxen sodium salts | The efficiency of drugs removal depends on the feed matrix: ultrapure water > tap water > secondary effluent > primary effluent. No drugs were detected in distillate, 99% DOC removal for both PE and SE, and no permeate flux decline for TW and SE. During PE treatment a significant flux decline (50–60%) was observed. Then, the PE should be pre-treated before the PMR. The high energetic consumption of MD must be considered. | [132] 2014 |

**Table 2.** *Cont.*

| PMR Configuration | Support | Photocatalyst | Pollutant | Main Results | Ref. Year |
|---|---|---|---|---|---|
| HPC + DCMD | polytetrafluoroethylene (PTFE) membrane | Ag/BiOBr | picrolonic acid | Ag/BiOBr photocatalyst mineralized PC into $CO_2$ and inorganic nitrogen species under visible light irradiation. Simultaneously MD permitted us to produce high-quality water as the distillate. The PTFE membrane stopped the passage of picrolonic acid and nitrogen species into the distillate. The high energetic consumption of MD must be considered. | [133] 2017 |

### 3.3. Visible Light as Energy Source

One important potential advantage that makes HPC a sustainable green approach is represented by the possibility of using solar energy as a renewable energy source.

As demonstrated by previous described works, the $TiO_2$ photocatalyst is the most utilized in PMRs [34,135,136]. This is because $TiO_2$ is characterized by a good photocatalytic activity, a relatively low rate of recombination of the electron-hole couples, a high photochemical stability, low cost, and toxicity [137,138]. Despite these advantages, it must be demonstrated that this material is able to use only less than about 5% of the energy of solar radiation. On this basis, the development of photocatalysts able to use visible light represents a key point in view of the large-scale application of PMR systems [139–141]. This trend is also reported in Figure 2, where a growing number of articles using visible light photocatalysts can be observed.

On this aspect, Athanasekou et al. [142] prepared photocatalytic ceramic UF membranes and tested them in the photocatalytic degradation of two azo-dyes, methylene blue (MB) and methyl orange (MO), under continuous dead-end flow conditions and near-UV/vis and visible light irradiation. The photocatalytic membranes were prepared by deposition on the external and internal surface of UF mono-channel monoliths by dip-coating three $TiO_2$ based photocatalysts: Nitrogen doped $TiO_2$ (N-$TiO_2$), graphene oxide doped $TiO_2$ (GO-$TiO_2$) and organic shell layered $TiO_2$. The results showed 57% and 27% degradation against MB and MO, respectively, by using the membrane coated with N-$TiO_2$ under UV irradiation, while 29% and 15% were obtained under visible light. These results, not encouraging from an environmental point of view, were caused by the used UF membrane support, which was not adequate for dye rejection, despite the photocatalyst deposition.

Carbamazepine degradation using an N-doped $TiO_2$ coated PMR was proposed and tested by Horovitz et al. in 2016 [143]. The photocatalytic membrane was prepared by coating a commercial α-$Al_2O_3$ photocatalytic membrane, characterized by 200 nm and 800 nm average pore size, with N-doped $TiO_2$ using a sol-gel technique. It was demonstrated that the permeability of the membrane after coating decreased by 50% and 12% for the 200 and 800 nm membrane support, respectively. The photocatalytic activity of the photocatalytic membranes was examined using a solar simulator as the light source. A significant gap in terms of degradation rates was observed by operating with the modality "flow through" the membrane and with the modality "flow tangential to" the surface of the membrane. In particular, recirculating the treated water through the photocatalytic membrane resulted in a significantly higher carbamazepine degradation. This result was mainly attributed to the so-called in-pore photocatalytic activity, due to increased contact of molecules with the active sites caused by the flow through the porous material. The results demonstrated an enhanced photocatalytic activity of N-doped $TiO_2$-coated membranes under UV wavelengths, in addition to activity under visible light, allowing more efficient

utilization of solar light. The authors also demonstrated that a disadvantage of coated PMRs in water treatment is the photocatalytic degradation controlled by diffusion of pollutants to the catalytic surface. The increase of mass transfer via an increase in water flux was found to be limited by membrane properties.

On the same topic, Gao et al. [144] studied the continuous removal of tetracycline using a PMR with $ZnIn_2S_4$ acting as both the adsorption and photocatalytic coating layer on a PVDF membrane. The photocatalytic membrane was prepared by deposition of $ZnIn_2S_4$ suspension on polyvinylidene fluoride (PVDF) membrane. A cold light source was selected as visible-light with the advantage of excellent optical properties and high luminous efficiency. Operating by recirculation of the effluent, the highest total organic carbon (TOC) removal efficiency of 57% was obtained after 3 h reaction time by operating under the following conditions: 1.88 mg cm$^{-2}$ photocatalyst, 84.06 L m$^{-2}$ h$^{-1}$ permeation flux, 50 mW cm$^{-2}$ light intensity. During continuous tests (i.e., under influent and effluent flux of 26.09 L m$^{-2}$ h$^{-1}$), a 50% removal efficiency was maintained during 24 h of photocatalytic reaction with 1.88 mg cm$^{-2}$ photocatalyst and 10 mg L$^{-1}$ initial concentration of tetracycline. By operating under the same conditions but with a lower drug concentration (100 μg L$^{-1}$), more than 92% removal efficiency was maintained for 36 h of continuous operation. These results demonstrated that the continuous influent and effluent operation mode might be more suitable for the final treatment of low concentration pollutants. The characterization of the membrane demonstrated that the surface and structure of PVDF membrane were not affected by the photocatalytic process, showing a good membrane stability.

Among various semiconductors, carbon materials, such as graphitized carbonitride (g-$C_3N_4$), have attracted the interest of researchers for its photocatalytic activity under visible light [145]. However, the photocatalytic activity of g-$C_3N_4$ in the visible range is limited due to its high photogenerated charge recombination rate. To improve the photocatalytic efficiency, it can be combined with other semiconductors such as $TiO_2$ [146].

The photocatalytic ability of a novel mesoporous graphitic carbon nitride/titanium dioxide (mpg-$C_3N_4$/$TiO_2$) nanocomposite in degrading the antibiotic sulfamethoxazole (SMX) under solar light was explored by Yu et al. [147] in the experimental set-up schematized in Figure 11. This novel nanomaterial was successfully synthesized and incorporated into a polysulfone (PSF) matrix to fabricate photocatalytic membranes. It was demonstrated that the pharmaceutically active compound SMX was transformed into 7 kinds of non-toxic and pharmaceutically inactive by-products by the PMR technology. SMX removal efficiency obtained by operating with the membrane named PSf-3 (with 1% mpg-$C_3N_4$/$TiO_2$ loading) was the highest (69%) during 30 h of consecutive irradiation. Meantime, the membrane structure was able to provide stable support with high integrity and flexibility after solar irradiation. Therefore, the results developed show that the prepared photocatalytic membrane has great potential to be applied in the water treatment industry.

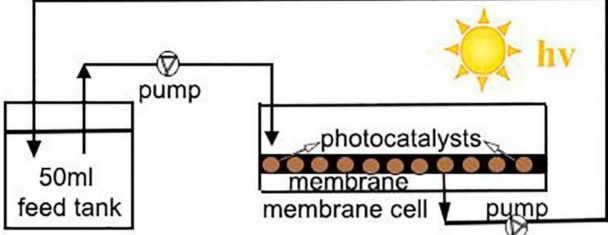

**Figure 11.** Schematic drawing of the PMR setup with a dead-end UF system with an active membrane area of 8.5 cm$^2$ used by Yu et al. Reprinted with permission from ref. [147]. Copyright © 2021 Elsevier B.V.

The application of PMR under visible light was very recently tested for the degradation of the drug diclofenac by using N-doped $TiO_2$ as the photocatalyst [148]. The performance of N-doped $TiO_2$ was evaluated using a submerged photocatalytic membrane

reactor (SMPR) with suspended N-doped TiO$_2$. The results indicated that higher initial concentration of the drug into the feed solution reduced the efficiency of the process, and that the addition of H$_2$O$_2$ enhanced system performance also in terms of increased degradation rate, as demonstrated by the fast disappearance of some degradation by-products.

A slightly different approach was proposed by Sheydaei et al. [149] for degradation of Reactive Orange 29 (RO29) as model organic pollutant. In this work, La-ZnO, Ho-ZnO and Ce-ZnO nanoparticles were synthesized by a sono-chemical method. These nanoparticles were used as photocatalysts under visible light in three reactor configurations: simple photocatalysis, sono-photocatalysis and sono-photocatalysis/membrane separation (SPMS). Various operating parameters of the synthesis procedure influenced the visible light photocatalytic activity of the prepared lanthanides-doped ZnO nanoparticles: the doping source, the mass ratio of doping source to the precursor of ZnO synthesis, the pH, the sonication and the calcination temperature and time. The optimum conditions for nanoparticles synthesis were: 8 wt% of cerium nitrate, pH 10, 1 h of sonication at 60 °C, 3 h of calcination at 300 °C. FE-SEM, EDS, XRD, PL and DRS analyses permitted us to identify the Ce-ZnO as the optimum catalyst. Then, the Ce-ZnO nanoparticles were used to remove Reactive Orange 29 (RO29) dye via sono-photocatalysis process under the visible light irradiation, in to determine the optimal chemical condition for improving the decolorization efficiency. Finally, a continuous flow visible light SPMS reactor was used in the presence of Ce-ZnO catalyst and polypropylene hollow fiber membrane for treatment of dye solution (Figure 12). In the best conditions, 97.84% of dye removal was achieved. GC-Mass, COD and TOC analyses permitted us to demonstrate the degradation and mineralization of RO29 using the SPMS process. Moreover, the prepared Ce-ZnO nanocomposite showed a favourable antibacterial behaviour against positive and negative bacteria.

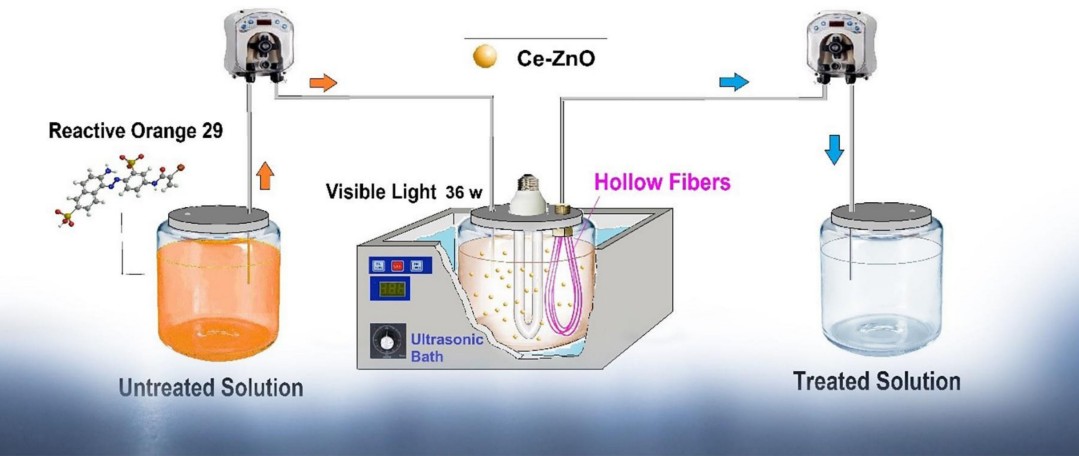

**Figure 12.** Schematic of SPMS reactor used by Sheydaei et al. Reprinted with permission from ref. [149]. Copyright © 2021 Elsevier B.V.

Visible-light photocatalysis was also tested in combination with membrane distillation by Huo et al. [150]. The authors tested the use of home-prepared BiOBr photocatalyst in the removal of Methyl Orange from aqueous solutions. High MO photodegradation efficiency was obtained by using the prepared hierarchical flower-like BiOBr microspheres. This result was ascribed to their specific surface area, their absorbance of visible-light and the lower recombination of photo-generated electron/hole couples. Moreover, a high quality permeate was obtained because both the organic pollutants and the catalyst were quantitatively maintained into the feed. A constant permeate flux was obtained during the photodegradation experiments, thus showing the absence of membrane fouling. Despite these encouraging results, the high energetic consumption of MD limited its coupling with HPC for water and wastewater treatment and no other works on this type of coupling was recently published on this topic. How we will report in the following section, this

interesting and promising coupling can become more interesting when the PMR is used not for photodegradation, but for more remunerative reactions for the synthesis of chemicals.

Mastropietro et al. [151], reported the use of $TiO_2/\alpha$-$Al_2O_3$ membranes that displayed self-cleaning properties permitting their reuse in successive catalytic runs without reduction in their photocatalytic activity for MB degradation. Photocatalytic tests were carried out by using UV or simulated solar light as irradiation source. The PMR was operated in cross-flow filtration mode, and tests were performed in continuous mode, with the feed/retentate and the permeate volumes being collected in separate chambers, or in batch, by recovering the permeate stream. The functionalized membranes were housed in a flat sheet membrane module equipped with a quartz window on the top side and then placed in a light exposure chamber. The MB aqueous solution ($10^{-5}$ M) was fed into the chamber with a pump flow rate of 14 mL min$^{-1}$ at a trans-membrane pressure set at 0.4 bar. The authors reported the complete MB degradation in only 40 min under solar light irradiation, when the $TiO_2/\alpha$-$Al_2O_3$ photocatalytic activity was synergistically combined with the $H_2O_2$-assisted oxidative reaction.

A new research topic which is arousing a lot of interest in recent years is the use of graphene (G) or G-based materials as additive or even as active photocatalyst for photocatalytic organic synthesis. Although the preliminary studies on the role of G in photocatalysis was focused on the improvement of the photocatalytic activity of semiconductor materials as active co-catalysts, subsequently it was demonstrated that some G-based materials possess intrinsic photocatalytic activity [152]. Moreover, when graphene based materials are prepared from biomass they are considered even more sustainable and their use can contribute to the green chemical industry compared to the metal-based photocatalysts [153]. Liu et al. [154] prepared a photocatalytic self-cleaning composite membrane obtained by combining LDH-based photocatalyst with graphene oxide. The layered double hydroxide @ graphitic nitrogen carbide (LDH@g-$C_3N_4$@PDA) composite photocatalysts were fabricated by dopamine modification method. Then, the LDH@g-$C_3N_4$@PDA and graphene oxide (GO) composites were assembled on PVDF membrane to construct the photocatalytic self-cleaning composite membrane. The presence of PDA can improve surface hydrophilicity, improve the removal rate of contaminants under visible light and enhance the self-cleaning or anti-contamination properties of the membrane [155]. Moreover, PDA has good absorption capacity for ultraviolet and visible light because it can promote the generation of photogenerated electrons and holes. The prepared PVDF/LDH@g-C3N4@PDA/GO composite membrane showed a rejection rates of methylene blue (MB), rhodamine B (RhB), gasoline, diesel, and petroleum ether which were 100%, 94.61%, 96.74%, 93.22%, and 92.35%, respectively. The authors reported a removal efficiency of 99.28% for MB while maintaining a flux of 397.14 L m$^{-2}$ h$^{-1}$ bar$^{-1}$ under 10 photocatalytic self-cleaning cycles [154].

In Table 3, a summary of the evolution of PMR configurations, working under visible light, described in this paragraph is reported. The results demonstrated that the development of proper photocatalyst and/or photocatalytic membranes permitted us to achieve enhanced photodegradation under UV radiation and photoactivity under visible light, allowing a more efficient utilization of solar light. Moreover, the use of solar light, characterized by a lower aggressiveness with respect to UV radiation, resulted in an increased membrane stability under irradiation. Despite these achievements, the lower mineralization achieved by using visible light, especially if it is necessary to remove recalcitrant pollutants, must be taken into correct consideration. Indeed, the photodegradation by-products could have a not negligible environmental impact.

**Table 3.** Summary of various PMR configurations working under visible light in water treatment and their main results.

| PMR Configuration | Support | Photocatalyst | Pollutant | Main Results | Ref. Year |
|---|---|---|---|---|---|
| HPC + MD | microporous hydrophobic flat sheet PTFE membrane | flower-like BiOBr microspheres | Methyl Orange | High efficiency for MO photodegradation. High quality permeate with constant flux. No membrane fouling. The high energetic consumption of MD limited its coupling with HPC for water and wastewater treatment. | [150] 2013 |
| Photocatalyst coated on the membrane | ceramic UF membranes | N-TiO$_2$ GO-TiO$_2$ organic shell layered TiO$_2$ | Methylene Blue (MB) and Methyl orange (MO) | 29% and 15% MB and MO degradations by using the membrane coated with N-TiO$_2$ under visible light irradiation. | [142] 2015 |
| Photocatalyst coated on the membrane | commercial α-Al$_2$O$_3$ photocatalytic membrane | N-TiO$_2$ | Carbamazepine | Degradation rates of "flow through" the membrane > degradation rates "flow tangential to" the surface of the membrane. Enhanced photoactivity of N-doped TiO$_2$-coated membranes under UV wavelengths, and activity under visible light. A disadvantage of coated PMRs: the photocatalytic degradation is controlled by pollutant diffusion to the catalytic surface. The increase of the mass transfer with increasing water flux was limited by membrane properties. | [143] 2016 |
| Photocatalyst deposited on the membrane | α-Al$_2$O$_3$ membranes | TiO$_2$ | MB | Complete MB degradation in only 40 min under solar light irradiation. | [155] 2017 |
| Photocatalyst coated on the membrane | PVDF membrane | ZnIn$_2$S$_4$ | Tetracycline | Removal efficiency > 92% was maintained for 36 h of continuous operation (under influent and effluent flux of 26.09 L m$^{-2}$ h$^{-1}$) with 100 μg L$^{-1}$ drug concentration. Good membrane stability: the surface and structure of PVDF membrane were not affected by the photocatalytic process. | [144] 2018 |
| Photocatalyst immobilized in the membrane | polysulfone membrane | novel mesoporous graphitic carbon nitride/titanium dioxide (mpg-C$_3$N$_4$/TiO$_2$) nanocomposite | Sulfamethoxazole | Sulfamethoxazole was degraded into 7 non-toxic and pharmaceutically inactive by-products by the PMR technology. Satisfactory sulfamethoxazole SMX removal efficiency was obtained by operating with the membrane named PSf-3 (with 1% mpg-C$_3$N$_4$/TiO$_2$ loading) for 30 h of consecutive irradiation. Good membrane stability: membrane provided a stable support with high integrity and flexibility after solar irradiation. The prepared photocatalytic membrane has a great potential to be applied in water treatment industry. | [147] 2018 |

<div align="center">

**Table 3.** *Cont.*

</div>

| PMR Configuration | Support | Photocatalyst | Pollutant | Main Results | Ref. Year |
|---|---|---|---|---|---|
| SMPR with suspended photocatalyst | polypropylene hollow fiber membrane | Ce-ZnO nanoparticles | Reactive Orange 29 | In the best conditions, 97.84% of dye removal was achieved in the continuous flow visible light SPMS reactor. GC-Mass, COD and TOC analyses demonstrated the degradation and mineralization of RO29. The Ce-ZnO nanocomposite showed a favorable antibacterial behavior against positive and negative bacteria. | [149] 2019 |
| SMPR with suspended photocatalyst | MF ceramic membrane | N-TiO$_2$ | Diclofenac | The efficiency of the photocatalytic process decreased by increasing the initial concentration of the drug while it was improved by adding H$_2$O$_2$. | [148] 2020 |
| PVDF/LDH@g-C3N4@PDA/GO composite membrane | PVDF | C$_3$N$_4$ | Methylene blue (MB), rhodamine b (RhB), gasoline, diesel, and petroleum | Rejection rates of methylene blue (MB), rhodamine B (RhB), gasoline, diesel, and petroleum ether were 100%, 94.61%, 96.74%, 93.22%, and 92.35%, respectively | [154] 2021 |

## 4. Evolution of PMRs in Reaction of Synthesis

HPC in membrane reactors has become very attractive in the last decade as an alternative method for the synthesis of organic compounds [6]. As shown in Figure 13 the number of articles on PMRs for partial oxidation reactions has a significant growth from years 2000 to 2020 as well as for reduction reactions shown in Figure 14.

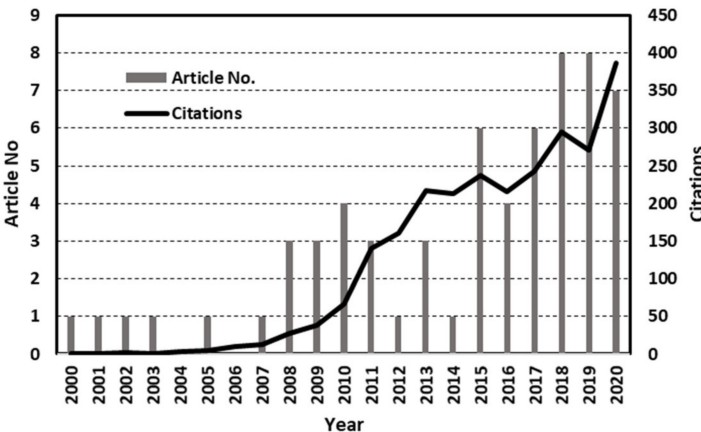

**Figure 13.** Number of articles (Article No.) on partial oxidation reactions and related citations regarding PMRs from the year 2000 to 2020 documented in the Scopus database (keywords: "photocatalytic membrane reactor" or "membrane photoreactor" or "photocatalytic membrane" and "partial oxidation").

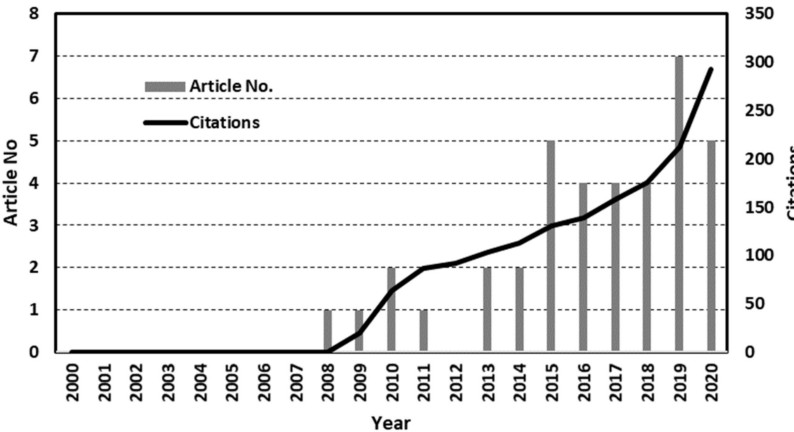

**Figure 14.** Number of articles (Article No.) on reduction reactions and related citations regarding PMRs from the year 2000 to 2020 documented in the Scopus database (keywords: "photocatalytic membrane reactor" or "membrane photoreactor" or "photocatalytic membrane" and "reduction" or "hydrogenation" and "synthesis").

The use of a PMR and the appropriate selection or modification of the photocatalyst and of some operating parameters during these 20 years has allowed for the increase in selectivity of photooxidation and photoreduction processes in comparison to conventional methods. In this context, the application of photocatalysis for partial oxidation of organic substrates has been mainly studied. This is because the most common semiconductors have VB edges more positive than oxidation potentials of most organic functional groups [156]. Photocatalytic reductions are less frequently found, since the reducing power of a CB electron is significantly lower than the oxidizing power of a VB hole [157].

The main purpose of combining a photocatalytic reaction of synthesis with a membrane process is the separation of the product(s) to improve yield and selectivity and the confinement of the photocatalyst into the reacting environment [1]. The photoreactors can have the photocatalyst in the liquid suspension and the photocatalyst immobilized in/on the membrane [44]. In the first group, the different geometries of the reactor and the different positions of the radiation source are important to determine the efficiency of the process, while in the second group, the light source must be positioned in front of the membrane active surface. In both groups, the key factor of the photocatalytic process is the photocatalyst, in which its structure and properties play a critical role in photocatalytic performance.

Based on the higher accessibility of the active surface area compared to an immobilized system, the PMRs with suspended photocatalysts have been most used in the literature, because they are more efficient than ones that use immobilized photocatalysts [104].

The selection of the appropriate operative conditions is of critical importance to obtain good performance of the PMRs finalized for practical applications. Thus, when developing a PMR, it is important to consider some parameters (photocatalyst, irradiation source) that influence the performance of the system.

### 4.1. PMRs Configuration in Reactions of Synthesis

One of the first problems when developing PMRs in the reactions of synthesis is the type of membrane material. A role of the membrane, when PMRs are applied for synthesis, is the separation of the product(s) from the reaction environment. Therefore, a high membrane permeability of the desired product is important to allow its rapid and efficient selective removal and recovery to improve selectivity and reaction yield. For example, an interesting reaction that requires high selectivity and good product extraction is the one-step production of phenol by direct hydroxylation of benzene. Indeed, the membrane in the PMR permits the product extraction thus avoiding excessive by-product formation. This is one of the most difficult oxidation reactions, because phenol is more reactive towards

oxidation than benzene, and substantial formation of by-products such as biphenyl and further oxidation compounds is usually found. In 2006, Molinari's group [158], in their first published work on this topic in a catalytic membrane reactor, demonstrated that process efficiency, quantified in terms of substrate and oxidant conversion to phenol, increases with the contact angle and the hydrophobic character of the membrane. Subsequently in 2009 [159], the authors demonstrated the possibility of using a photocatalytic hybrid system in which the selective photocatalytic reaction and the separation of the product of interest occurred in one step by using a polypropylene membrane to separate the organic phase from the aqueous environment according to their previous work [158]. Recently, the hydrophobic polypropylene membrane was used successfully by Molinari's group also for photocatalytic reduction in acetophenone in PMR. Different methods for the substrate addition in the membrane photoreactor were tested, obtaining the best performance using acetophenone as both solvent and reactant (substrate). By operating in this way, 21.91% extraction percentage in the organic phase (Q%) and 4.44 mg $g^{-1}$ $h^{-1}$ productivity vs. 2.96 mg $g^{-1}$ $h^{-1}$ of the PMR compared to the batch photoreactor were obtained [160].

Even though the hydrophobic polypropylene membrane gave these good results in the PMR [161], this membrane material does not meet the fouling resistance criteria and for this reason it can be prone to fouling, in particular to biofouling. Membrane fouling is usually due to a foulant adhesion/deposition and thermodynamic filtration resistance of the gel/cake layer. The main disadvantage of membrane fouling consists in the block of membrane pores caused by the increase of foulants deposition on the membrane surface that causes a decrease in flux through the membrane and membrane life, which involves, therefore, a cost grows of membrane technology. To avoid this problem, in recent years some researchers have studied the fabrication of membranes with ability to degrade foulants [162]. For example, biofouling can be reduced by exploiting photocatalysis to pre-treat the feed solution prior to membrane filtration to eliminate bacteria.

Very recently, Lin et al. [161] studied the use of S-doped g-$C_3N_4$ nanosheet as a photocatalyst for both water splitting and biofouling reduction. They studied the photocatalytic $H_2$ and $O_2$ generation by using this photocatalyst in combination with Ru/SrTiO$_3$: Rh with the addition of [Co(bpy)$_3$]$^{3+}/^{2+}$ as electron mediator to improve the charge transfer in a Z-scheme system. The $H_2$ and $O_2$ evolution rates in the system were 24.6 and 14.5 $\mu mol^{-1}$ $h^{-1}$, respectively. Furthermore, S-doped g-$C_3N_4$ was incubated with a solution of *Escherichia coli* to verify its antibacterial effect. The results showed that S-doped g-$C_3N_4$ has a high activity in biofouling reduction on a membrane.

In 2012, Augugliaro et al. [163], tested a PMR obtained by coupling HPC and pervaporation (PV) in the photocatalytic oxidation of trans-ferulic acid to vanillin (VA). The reported results, showed that use of a nonporous PEBAX 2533 (trade name of polyether-polyamide block copolymers) membrane, permitted a high permeability toward VA (transmembrane flux about 3.31 $g_{VA}$ $h^{-1}$ $m^{-2}$), improving the removal of the product from the irradiated suspension limiting its subsequent oxidation, thus increasing process selectivity. In this type of system, another important parameter influencing its performance is membrane thickness. In 2014, Camera-Roda et al. [164], studying VA pervaporation with PEBAX membranes, demonstrated that increasing membrane thickness allowed the improvement of the enrichment factor of VA (VA concentration in the condensed permeate/VA concentration in the feed) because the resistance to VA permeation remains low while the resistance to water permeation increases. Although pervaporation has been considered a very attractive membrane process for recovering vanillin in a membrane reactor, dialysis with the same polymeric PEBAX membranes has several advantages. The most important is that dialysis does not need the evaporation of the permeating species, resulting in a lower energy consumption. Moreover, the low volatility of vanillin does not limit the permeation. The most important effect is that the permeate flux obtained by using dialysis is at least one order of magnitude higher than in pervaporation. Furthermore, dialysis is effective also at ambient temperature and this aspect makes it suitable also for the bioreactions that cannot withstand high temperatures. In 2020, Camera-Roda et al. [59] used a PMR obtained by

coupling dialysis with HPC for the photocatalytic partial oxidation of ferulic acid in an aqueous solution at ambient temperature. The rate of vanillin formation was improved, compared to other reactor configurations, because intermediate compounds permeated from the reacting solution and did not hinder the reaction, while ferulic acid permeated in the opposite direction to partially supply the reactor with the substrate. The experimental PMR with $TiO_2$ suspended in the aqueous solution is schematized in Figure 15. The integration of the HPC process with the MS step was obtained by continuously recirculating the reacting solution (Liquid 1) through the reactor and the membrane modulus.

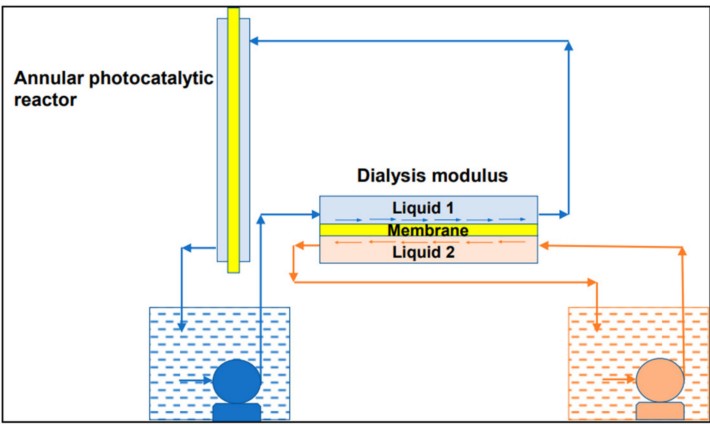

**Figure 15.** Scheme of the experimental set-up obtained by continuously recirculating the reacting solution (Liquid 1) through the photocatalytic reactor and the membrane modulus. Reprinted with permission from ref. [59]. Copyright © 2021 MDPI.

The authors reported that by using this system, the amount of vanillin produced in 5 h in the PMR was more than one-third higher than in the PR without dialysis.

### 4.1.1. Photocatalyst Immobilized in/on the Membrane in Reactions of Synthesis

An ideal photocatalyst must take the following qualities, i.e., inexpensive, excellent stability during photoreaction, non-toxic to the ecosystem, highly selective towards the targeted material and recoverable [5]. The photocatalyst in immobilized form is easy to be recovered/reused several times, but the photocatalyst must be stable with a strong entrapment onto the membrane materials [165].

Several strategies in place to secure the catalyst onto the membrane material are: establishing a chemical bond such as covalent bindings, electrostatic interaction, Van der Waals (hydrogen bond), or its encapsulation to another chemical that could promote bonding [166]. Another way is the encapsulation of the catalyst immobilizing it inside the membrane matrix to obtain a mixed matrix membrane [167]. In the following, some photocatalysts and the type of immobilization on membranes is reported.

Recently, graphene quantum dots (GQDs) have attracted growing interest thanks to their high quantum effect and large specific surface areas [168]. In 2016, Liu et al. [169], proposed the preparation of a sandwich GQDs–$Cu_2O$/BPM with GQDs–$Cu_2O$ catalyst inside the interlayer. The results showed that GQDs–$Cu_2O$ decreased the impedances of the membrane under sunlight irradiation and that GQDs–$Cu_2O$/BPM minimized the formation of pH gradient.

In 2018, Zhao et al. [170] studied a composite membrane (C-doped $Cr_2O_3$/NaY) supported on stainless steel mesh and tested it in the photocatalytic oxidation of cyclohexane. They prepared C-doped $Cr_2O_3$ photocatalyst, using a chromium-containing MOF as precursor, on the upper surface of NaY zeolite membrane. The obtained composite membrane presented three layers: i) the stainless-steel mesh, acting as supported carrier (on the bottom), ii) the NaY membrane, used as adsorbent (in the middle), iii) and the photocatalyst C-doped $Cr_2O_3$ (on the top). With this system they obtained a selectivity to KA oil (a mixture of cyclohexanone (K) and cyclohexanol (A)) up to 99.73%, coupled with

an unsatisfactory conversion of cyclohexane (0.93%). The NaY zeolite membrane, with high specific surface area, suitable pore size, uniform pore size distribution and polarity, had the dual role of capturing the product and avoiding its over-oxidation, thus resulting in an improvement of the yield of KA oil.

The use of immobilized photocatalyst on a membrane can show various advantages and disadvantages. Diaz-Torres et al. [171] tested the hydrogen generation properties of $ZnAl_2O_4$ (ZAO) powders, synthesized by a combustion method, which produced carbon dots (C-dots) on the ZAO surface, alone or incorporated into a polyacrylate matrix to form a photocatalytic membrane (named PAZO) which was subsequently attached to a flexible graphene composite (FGC) to form a FGC/PAZO (GAZO) composite. The authors reported that although the hydrogen generation rate under UV irradiation was lower by using GAZO composite ($\approx$38% less) than the best ZAO powder (annealed at 700 °C), the GAZO composite could be attached easily in the inner wall of photocatalytic reactors which facilitates its removal after hydrogen production, this advantage is not possible by using photocatalytic powders.

4.1.2. $CO_2$ Conversion

In recent years, many studies on photocatalysts immobilized in/on a membrane concerned the photocatalytic conversion of carbon dioxide ($CO_2$) because the increased interest on its transformation to useful products has a high potential to address the adverse environmental impact of global warming [172]. Some authors reported the use of $TiO_2$ nanoparticle immobilized in the membrane structure for $CO_2$ photocatalytic reduction. In this regard, Cheng et al. in 2016 [173] applied an optofluidic microreactor for $CO_2$ photoreduction to methanol by using a $TiO_2$/carbon paper composite membrane. The authors reported a methanol production yield of 111 μmol $g_{cat}^{-1}$. To enhance $CO_2$ photoconversion, Maina et al. [174] used $TiO_2$ and Cu-$TiO_2$ as photocatalysts within zeolitic imidazolate framework (ZIF 8) membranes. They reported an improved methanol yield by 70% and CO yield by 233% incorporating Cu-$TiO_2$ nanoparticles with ZIF-8 membranes. Recently Baniamer et al. [172], reported the use of a two-layer photocatalytic membranes fabricated from a porous polyethersulfone-$TiO_2$ (PES-$TiO_2$) as a photocatalytic layer and a thin nonporous layer of poly-ether-block-amide (PEBAX-1657) as a selective layer to obtain simultaneous separation and conversion of $CO_2$. The authors obtained the high methanol production yield of about 697 μmol $g_{cat}^{-1}$ in the presence of water at 5 wt% of $TiO_2$ nanoparticle contents, 3 mL $min^{-1}$ of water flow rate and 8.84 W $cm^{-2}$ of light power.

Another type of semiconductor that has recently attracted much attention is graphite carbon nitride (g-$C_3N_4$), which possesses a two-dimensional (2D) nanosheet structure like graphene. g-$C_3N_4$ exhibited many useful properties with applications in many fields, such as materials for membrane separation, photocatalysis, and electronic devices [175]. The basic skeleton structure of g-$C_3N_4$ consists of tri-s-triazine units connected with tertiary amino groups, which owns regularly distributed triangular water-selective permeation nanopores throughout the entire laminar structure [176]. Moreover, the spacers between the g-$C_3N_4$ nanosheets, also provide nanochannels for water transport while bigger molecules are retained [177]. g-$C_3N_4$ shows other advantages for $CO_2$ photo-reduction because it is rich of N basic sites, which favors the $CO_2$ adsorption step.

In 2019, Brunetti et al. [175], investigated the photocatalytic $CO_2$ reduction that was carried out in a continuous photocatalytic reactor with an exfoliated $C_3N_4$-$TiO_2$ photocatalyst embedded in a dense Nafion matrix, irradiated by UV light. The authors reported that MeOH production increased with the $TiO_2$ content in the catalytic membrane, increasing from 17.9 when only $C_3N_4$ was embedded into Nafion membrane to 45 μmol $g_{catalyst}^{-1}$ $h^{-1}$ for 100% of $TiO_2$. This might be also attributed to the heterojunction formation of $C_3N_4$ based materials, which usually enhances photocatalytic performance to $CO_2$ conversion [178].

### 4.1.3. Magnetic Materials and Optical Fiber

A more recent method for recovering the photocatalyst from water is the use of new materials such as magnetic composites. In the last decade, the interest in developing new materials, including graphene-based semiconductors, has enhanced because of their adsorptive ability towards pharmaceuticals [179]. The continuous advancements in PMR research permitted the development of various materials, e.g., adsorbents incorporated in membrane technologies or photocatalysts combined with magnetic material and coated on optical fibers. For example, to reduce $CO_2$ with $H_2O$ to fuels under UVA artificial light and concentrated natural sunlight, Nguyen al. used an optical-fiber reactor prepared by coating it with a gel-derived $TiO_2$–$SiO_2$ mixed oxide-based photocatalyst [180]. In this system the visible light absorption and product selectivity were influenced by the insert of Fe atom into the $TiO_2$–$SiO_2$ lattice during the sol–gel process. The authors reported that by using Cu–Fe/$TiO_2$ the main product was ethylene with the quantum yield of 0.0235%, while by using Cu–Fe/$TiO_2$–$SiO_2$ as photocatalyst was favored the generation of methane as primary product with a quantum yield of 0.05%, both under UVA irradiation. Only methane was produced by using both $TiO_2$–$SiO_2$ and Cu–Fe/$TiO_2$–$SiO_2$ photocatalysts under natural sunlight with the production rates of 0.177 and 0.279 mmol $g_{cat}^{-1}$ $h^{-1}$, respectively.

Cheng et al. [181] studied the photocatalytic reduction in $CO_2$ in an optofluidic planar microreactor irradiated by a 100 W LED (365 nm). High purity $CO_2$ (99.99%) was continuously fed to an aqueous alkaline solution for 1 h with the dual scope of removing dissolved oxygen and saturating it with $CO_2$. Then, the $CO_2$ saturated aqueous solution was fed to the microreactor by a syringe pump. The microreactor was constituted by a porous $TiO_2$ film coated glass as the bottom substrate, which was formed by the wet spray method, and by a transparent rectangular reaction chamber. The results showed that both the methanol concentration and yield was improved at high light intensity and NaOH concentration. The best performance of the system (methanol yield of 454.6 mmol $g_{cat}^{-1}$ $h^{-1}$) was obtained by improving the catalyst loading, using a liquid flow rate of 50 mL $min^{-1}$, 0.2 M NaOH, and 8 mW $cm^{-2}$ light intensity.

To enhance the visible-light responsive $CO_2$ photoreduction in an optofluidic membrane microreactor, Chen et al. [182] studied a novel mesoporous CdS/$TiO_2$/SBA-15@carbon paper composite membrane. The $CO_2$ photoreduction system mainly consisted of five components: a syringe pump, a $CO_2$ gas cylinder, an optofluidic membrane microreactor, a simulated sunlight source and a collection vessel as showed in Figure 16.

Experimental results showed that the optofluidic membrane microreactor with mesoporous CdS/$TiO_2$/SBA-15@carbon paper composite membrane yielded much better performance than did the one without the mesoporous SBA-15. In addition, it was found that the methanol concentration and yield firstly increased and then decreased with increasing the liquid flow rate. The authors reported a maximum methanol yield of about 1022 mole $g_{cat}^{-1} \cdot h^{-1}$ obtained by using CdS/20 wt% $TiO_2$/SBA-15 at 0.4 M NaOH concentration, which was nearly 4 times higher than CdS/$TiO_2$. The higher NaOH concentration is beneficial for the $CO_2$ photoreduction and the incorporation of SBA-15 shows the advantageous performance than conventional CdS/$TiO_2$.

Summarizing, the results reported in Section 4.1 show that to use PMRs for synthetic purposes it is important to choose a membrane with a high permeability toward the desired product, thus obtaining its prompt removal from the reacting environment and limiting its subsequent reduction/oxidation and increasing process selectivity. Regarding the possibility of using the entrapped photocatalyst, the results evidence that this system configuration results in easier photocatalyst recovery and reuse, but sometimes a lower productivity with respect to slurry PMR is obtained.

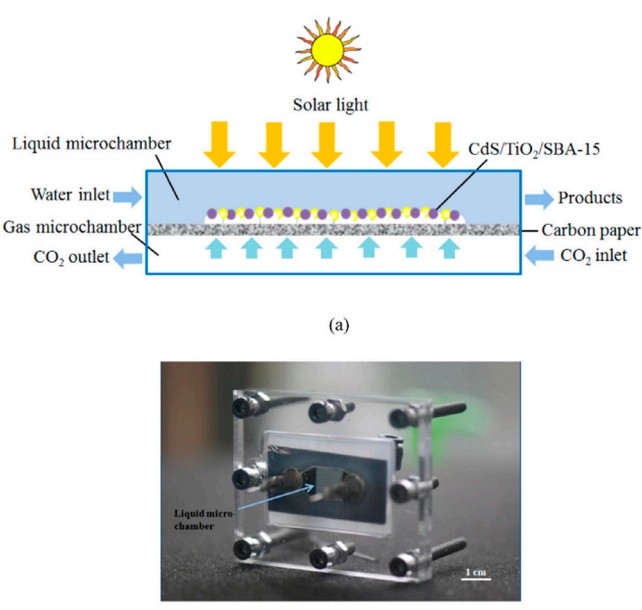

**Figure 16.** (**a**) Schematic and (**b**) photo of the optofluidic membrane microreactor. Reprinted with permission from ref. [182]. Copyright © 2021 Elsevier B.V.

### 4.2. Visible Light as Energy Source in Reactions of Synthesis

The increased interest in visible light irradiation source over UV source, demonstrated also in paragraph 3.3, is justified by different reasons. An important point that must be taken into consideration is the possible deterioration of the membrane by UV irradiation that can limit the membrane efficiency [166] and can influence also by-products formation. Indeed, the excessive energy input in the reacting environment caused by high-energy UV-light often induces an increase of by-products formation due to higher generation of reducing and/or oxidizing agents. One advantage of using visible light as irradiation source is the possibility to increase the selectivity values due to the lower photon energy required to activate the photocatalyst [183,184]. During the last decade the effort of scientists increased to develop new photocatalysts or semiconductors combination or their modification to improve the photocatalytic activity, to limit the electron-hole recombination and to improve the absorption on the visible light range [4].

Comparing productivity or selectivity values from some different works on photocatalytic synthesis in PMRs under UV or visible light, the bests results were obtained under visible light irradiation. For example, Molinari et al. [160], reported the comparison of commercial $TiO_2$ and homemade $Pd/TiO_2$ photocatalytic activity under UV and visible light for the photocatalytic hydrogenation of acetophenone (AP) to phenyl ethanol. The photocatalytic tests were conducted by using water as solvent and formic acid as electron and hydrogen donor in batch and membrane reactors. The authors compared various methods for adding the substrate in the membrane photoreactor. The best performance was achieved by using AP as both solvent and reactant (substrate) obtaining a productivity of 4.44 mg g$^{-1}$ h$^{-1}$ vs. 2.96 mg g$^{-1}$ h$^{-1}$ PMR vs. batch reactor. The enhanced productivity of PMR with respect to the batch system was due to the simultaneous extraction of the produced phenyl ethanol in the organic phase that shifted the hydrogenation reaction forward to the product. Moreover, the extraction of phenyl ethanol from the reactive phase permitted us to prevent subsequent over-reduction in the extracted product, thus improving the selectivity of the overall process. To improve the visible light activity of the photocatalyst, titanium dioxide was doped with Pd obtaining a productivity value five times higher by using $Pd/TiO_2$ than pure $TiO_2$ (productivity 22.0 mg g$^{-1}$ h$^{-1}$ vs. 4.44 mg g$^{-1}$ h$^{-1}$). Figure 17 shows the phenyl ethanol extraction in the organic phase by using $Pd/TiO_2$ and $TiO_2$ (under UV and visible light).

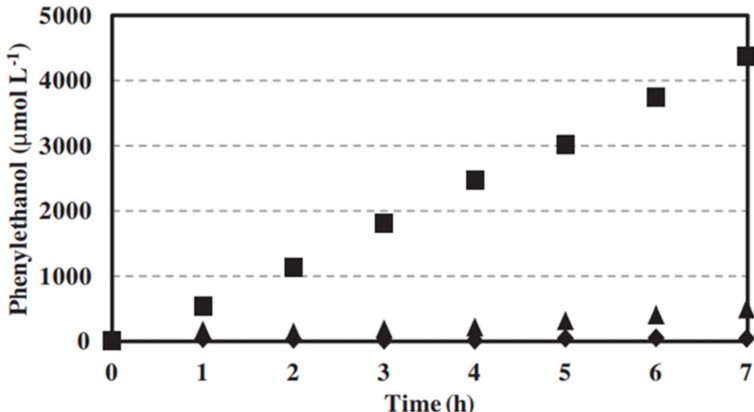

**Figure 17.** Phenyl ethanol extraction in the organic phase using $Pd/TiO_2$ and $TiO_2$ (under UV and visible light). Reprinted with permission from ref. [160]. Copyright © 2021 Elsevier B.V.

Recently, Lavorato et al. [21] prepared and tested, for the photocatalytic hydrogenation of AP in batch and in a membrane reactor, various photocatalysts prepared with $TiO_2$ and faujasite (FAU): $TiO_2$-loaded faujasite (FAU) zeolite and $Pd/TiO_2/FAU$. Preliminary photocatalytic tests were conducted under UV light and the best identified photocatalyst was named TF10P. Subsequently, this photocatalyst was doped with Pd (Pd_TF10P) obtaining a photocatalyst active under visible light. The productivity obtained in the PMR was higher by using Pd_TF10P than $Pd/TiO_2$ under visible light irradiation (productivity 99.6 mg $g_{TiO_2}^{-1}$ $h^{-1}$ vs. 22 mg $g_{TiO_2}^{-1}$ $h^{-1}$) with an extraction percentage of phenyl ethanol of around 25%.

To control the pore size and the doping level, Su et al. [185] studied the fabrication of Al- and Zn-doped $TiO_2$ nanotubes by atomic layer deposition (ALD) combined with polycarbonate (PC) membrane as the template. The bilayers were alternately deposited on the PC membrane template by ALD with various cyclic sequences. Zn doped $TiO_2$ nanotubes with optimal doping levels (Zn doping ratio 0.01) exhibited 6 times higher photocatalytic hydrogen production rate than pure $TiO_2$ under irradiation with a 150 W Xe lamp (Figure 18). The results suggested that Zn-doped $TiO_2$ nanotubes contain a certain amount of electron trapped $Ti^{3+}$ surface states and surface oxygen vacancies. Both of which contribute significantly to visible light absorption and photocatalytic performance.

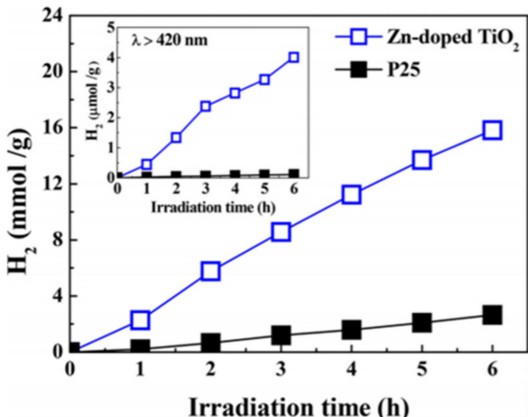

**Figure 18.** Hydrogen evolution from water splitting by Zn-doped $TiO_2$ nanotubes. The inset shows hydrogen evolution under visible light Reprinted with permission from ref. [185]. Copyright © 2021 American Chemical Society.

Zhang et al. [186], reported high selectivity values for the oxidation of a series of aromatic alcohols under visible light irradiation, by using a system constituted by a dye-

sensitized anatase $TiO_2$-TEMPO (2,2,6,6-tetramethylpiperidinyloxyl) nanoparticles. The selectivity values obtained with this system were higher than 93% for oxidation of some aromatic alcohols and of 98% for benzyl alcohols.

Other semiconductors, which attracted great interest on their use as photocatalysts, are metal-organic frameworks (MOFs). They consist of metal clusters interconnected with organic linkers (e.g., MOF-5, UiO-66(Zr), ZIF-8, MIL-125(Ti), etc.) [187]. These semiconductors showed a limited efficiency under solar light illumination, but the modification of their organic linkers or metal centers can overcome this limitation making their use possible under visible light. Moreover, in recent years, researchers are increasingly interested in using sustainable materials to obtain bio-membranes. For example, chitosan is an attractive natural biopolymer that can be used to synthesize membranes. Furthermore, the abundant presence of amine and carboxyl groups, are favorable for $CO_2$ adsorption. In 2019, Zhao et al. [188] studied the photocatalytic conversion of $CO_2$ under visible light irradiation and prepared a membrane using chitosan as the carrier. They obtained a membrane matrix named $CdS/NH_2$-UiO-66 hybrid membrane (where UiO-66 is a metal–organic framework (MOF)) obtaining high surface area and thermal stability. The synergistic activity obtained by incorporating MOFs and semiconductors into membranes, improved the $CO_2$ photocatalytic reduction reaction because this hybrid membrane accelerated the electrons transfer limiting the recombination of electron–hole pairs.

Summarizing, the results obtained by using visible light as energy source in reactions of synthesis demonstrated that by-products formation, which is frequently induced by using high-energy UV-light, can be limited by using visible light. Then, the use of visible light active photocatalysts is highly recommended when HPC is used for synthetic purposes, since this "low-energy" irradiation permits us to obtain higher process productivity. The modification of photocatalysts to achieve activity under visible light makes HPC a sustainable green approach, giving the possibility of using the solar energy as a renewable energy source.

The main results described in Section 4 and their evolution over time are summarized in Table 4.

**Table 4.** Summary of some PMRs tested in organic synthesis and the main results.

| PMR or Membrane Type | Photocatalyst | Application | Main Results | Ref. Year |
|---|---|---|---|---|
| Optical-fiber reactor under sunlight | $TiO_2$–$SiO_2$ | $CO_2$ reduction | Methane production rates of 0.177 mmol $g_{cat}^{-1}$ $h^{-1}$ | [180] 2008 |
| Optical-fiber reactor under sun light | Cu–Fe/$TiO_2$–$SiO_2$ | $CO_2$ reduction | Methane production rates 0.279 mmol $g_{cat}^{-1}$ $h^{-1}$ | [180] 2008 |
| Polypropylene | $TiO_2$ | Benzene oxidation to phenol | Extraction percentage of around 24% | [159] 2009 |
| Nonporous PEBAX 2533 by coupling HPC and PV | $TiO_2$ | Photocatalytic oxidation of trans-ferulic acid to vanillin | High permeability toward VA (transmembrane flux about 3.31 $g_{VA}$ $h^{-1}$ $m^2$) | [163] 2012 |
| PEBAX membrane pervaporation | $TiO_2$ | Synthesis of vanillin | Enrichment factor of VA improved | [164] 2014 |
| Polypropylene | $TiO_2$ and Pd/$TiO_2$ | Hydrogenation of acetophenone | Q% equal to 21.91%, productivity 4.44 mg $g^{-1}$ $h^{-1}$ vs. 2.96 mg $g^{-1}$ $h^{-1}$ of PMR vs. batch reactor | [160] 2015 |
| GQDs–$Cu_2O$/BPM with catalyst inside the interlayer | GQDs–$Cu_2O$ | Water splitting | Membrane impedances and pH gradient formation decreased | [169] 2016 |

**Table 4.** *Cont.*

| PMR or Membrane Type | Photocatalyst | Application | Main Results | Ref. Year |
|---|---|---|---|---|
| Optofluidic microreactor with $TiO_2$/carbon paper composite membrane | $TiO_2$ | $CO_2$ photoreduction | Methanol production yield of 111 $\mu$mol $g_{cat}^{-1}$. | [173] (2016) |
| Photocatalyst within zeolitic imidazolate framework (ZIF 8) | $TiO_2$ and Cu-$TiO_2$ | $CO_2$ photoconversion | Methanol yield by 70% and CO yield by 233% | [174] (2017) |
| Polypropylene | Pd/$TiO_2$/FAU | Hydrogenation of acetophenone | Productivity 99.6 mg $g_{TiO_2}^{-1}$ $h^{-1}$ vs. 22 mg $g_{TiO_2}^{-1}$ $h^{-1}$ of PMR vs. batch reactor under visible light, Q% around 25% | [21] 2017 |
| Optofluidic planar microreactor irradiated by a 100 W LED (365 nm) | $TiO_2$ film | $CO_2$ reduction | methanol yield 454.6 mmol $g_{cat}^{-1}$ $h^{-1}$ | [181] 2017 |
| Optofluidic membrane microreactor with simulated sun light | CdS/20 wt% $TiO_2$/SBA-15 | $CO_2$ reduction | 1022l mole $g_{cat}^{-1}$ $h^{-1}$ obtained by using CdS/20 wt% $TiO_2$/SBA-15 at 0.4 M NaOH concentration, | [182] 2017 |
| Membrane matrix | CdS/$NH_2$-UiO-66 | $CO_2$ reduction | improved $CO_2$ photocatalytic reduction under visible light irradiation (521.9 mmol $g^{-1}$ of CO produced) | [188] 2018 |
| Polycarbonate membrane | Zn doped $TiO_2$ nanotubes | Hydrogen production | 6 times higher photocatalytic hydrogen production rate than pure $TiO_2$ | [185] 2018 |
| Composite membrane supported on stainless steel mesh | C-doped $Cr_2O_3$/NaY | Cyclohexane oxidation | Selectivity to KA oil 99.73%, conversion efficiency of cyclohexane 0.93%. | [170] 2018 |
| Continuous photocatalytic reactor irradiated by UV light | Exfoliated $C_3N_4$-$TiO_2$ photo-catalyst embedded in a dense Nafion matrix. | $CO_2$ reduction | MeOH production 45 $\mu$mol $g_{catalyst}^{-1}$ $h^{-1}$. | [175] 2019 |
| Water splitting and biofouling reduction | S-doped g-$C_3N_4$ | Water splitting | $H_2$ and $O_2$ evolution rates in the system were 24.6 and 14.5 $\mu$mol$^{-1}$ $h^{-1}$. Biofouling reduction. | [161] 2020 |
| Photocatalytic membrane reactor (dialysis) | $TiO_2$ | Photocatalytic oxidation of trans-ferulic acid to vanillin | The total amount of vanillin produced after 5 h in the membrane reactor was more than one-third higher than in the photocatalytic reactor without dialysis. | [59] 2020 |
| GAZO composite | $ZnAl_2O_4$ | Hydrogen generation | Hydrogen generation rates of 4640 and 2860 $\mu$mol $g^{-1}$ $h^{-1}$ were obtained for ZAO powder and GAZO composite, respectively. | [171] 2020 |
| Two-layer photocatalytic membranes: polyethersulfone-$TiO_2$ (PES-$TiO_2$) and poly-ether-block-amide (PEBAX-1657) | $TiO_2$ | Conversion of $CO_2$ | Methanol production yield about 697 $\mu$mol $g_{cat}^{-1}$ $h^{-1}$ in the presence of water at 5 wt% of $TiO_2$ nanoparticle contents, 3 mL min$^{-1}$ of water flow rate and 8.84 W cm$^{-2}$ of light power. | [172] 2021 |

## 5. Conclusions and Future Trends

The evolution of the scientific knowledge of photocatalytic membrane reactor (PMR) technology, in twenty years of research, has made significant progress. The initial problems discussed previously and how they have been solved/reduced over time have improved the performance of PMRs. In the initial research period, most of the studies applied to PMRs concerned only the use of UV light, and in general they were directed towards water and wastewater treatment because UV driven reactions are unselective and they can totally degrade (mineralize) pollutants into innocuous substances. On this topic, around 750 studies were registered in the Scopus database in the period from 2000–2020, compared to 63 studies on the use of PMR for synthesis. In parallel, the interest of the scientific community in PMR technology has exponentially increased.

The results reported in this review evidence the feasibility of using PMR for water and wastewater treatment. Some important aspects to be considered are the type of PMR configuration. By using the suspended photocatalyst rather than the immobilized photocatalyst, it is possible to achieve higher efficiency thanks to the larger active surface area, which guarantees better contact between the photocatalyst and the substrate. Split type PMRs, with two separate vessels for HPC and MS, is the most promising PMR configuration in view of its large-scale application, since the two coupled processes can be separately implemented and optimized. The configuration named submerged PMR (SPMR) with air bubbling and back-flushing seems the most suitable for water and wastewater treatment because it permits us to limit membrane fouling and light scattering from photocatalyst particles. Regarding the possibility of using solar light to achieve water treatment through pollutant mineralization, the lower mineralization, achieved by using visible light, especially if it is necessary to remove recalcitrant pollutants, must be taken into consideration. Recently, the modification of photocatalyst, the innovation in PMR configuration, and the improved absorption of visible light have increased the application of PMRs also towards reduction and the partial oxidation of organics.

In recent, interest in solar-driven photocatalytic conversion has become more attractive because the use of visible light as an irradiation source allowed us to attain improved selectivity towards reduction and partial oxidation products and because it permits us to use solar energy in a clean and effective way for conducting chemical reactions. In this context, the configuration of a photocatalytic membrane (with an immobilized photocatalyst) seems more suitable since it results in easier photocatalyst recovery and reuse. An important aspect to be taken into consideration is the choice of the membrane, which must be characterized by high permeability toward the desired product, allowing for its prompt removal from the reaction environment.

The advantages discovered during these last two decades of research on PMRs applied to the partial oxidation and reduction in organics are: (i) decreased degradation rates of polymeric membranes, thus elongating their lifetime by using the visible light as an irradiation source; (ii) photocatalyst recovery can be improved by using innovative materials in the preparation of photocatalyst composites and coating semiconductors on optical fibers.

In conclusion, PMR technology seems quite mature for applications in water and wastewater treatments by taking advantage of the knowledge gained about SPMRs. The progress on photovoltaic technology (conversion of sunlight into electrical energy) and use of LED lamps (UV and/or visible) could help identify the best choice of photocatalyst for a specific reaction (degradation or synthesis) and simplify PMR design to tie the photocatalyst and the optimal coupling of the photocatalyst with the membrane together. It is thus expected that within a short time, we will see an increase in industrial interest in such systems.

**Author Contributions:** C.L. and P.A. elaborated the literature overview and wrote the first draft; C.L., P.A. and R.M. organized the paper; C.L., P.A. and R.M. revised the paper. All authors have read and agreed to the published version of the manuscript.

**Funding:** This research received no external funding.

**Data Availability Statement:** In this section, please provide details regarding where data supporting reported results can be found, including links to publicly archived datasets analyzed or generated during the study. Please refer to suggested Data Availability Statements in section "MDPI Research Data Policies" at https://www.mdpi.com/ethics. You might choose to exclude this statement if the study did not report any data.

**Conflicts of Interest:** The authors declare no conflict of interest.

## Abbreviations

| | |
|---|---|
| ALD | Atomic layer deposition |
| AOP | Advanced oxidation process |
| AR1 | Acid red 1 |
| AR4 | Acid red 4 |
| AR18 | Acid red 18 |
| Article No. | Article number |
| AY36 | Acid yellow 36 |
| CA | Cellulose acetate |
| CB | Conduction band |
| CNTs | Carbon nanotubes |
| 4-CP | 4-Chlorophenol |
| CTA | Cellulose triacetate |
| DCF | Diclofenac |
| DCMD | Direct contact membrane distillation |
| DG99 | Direct green 99 |
| DRS | Differential reflectance spectroscopy |
| 2,4-DHBA | 2,4-Dihydroxybenzoic acid |
| EDS | Energy dispersive X-ray spectroscopy |
| $E_g$ | Band gap of energy |
| FA | Fulvic acid |
| FE-SEM | Field emission scanning electron microscopy |
| G | Graphene |
| g-$C_3N_4$ | Graphite carbon nitride |
| GO | Graphene oxide |
| GODs | Graphene quantum dots |
| GO-$TiO_2$ | Graphene oxide doped $TiO_2$ |
| HFM | Hollow fiber membrane |
| HPC | Heterogeneous photocatalysis |
| IBU | Ibuprofen |
| IR | Infrared |
| MB | Methylene blue |
| MD | Membrane distillation |
| MF | Microfiltration |
| MOFs | Metal organic frameworks |
| MO | Methyl orange |
| MR | Membrane reactor |
| MS | Membrane separation |
| NAP | Naproxen |
| NF | Nanofiltration |
| NIR | Near infrared |
| NMP | N-methyl-2-pyrrolidone |
| 4-NP | 4-Nitrophenol |
| NPs | Photocatalyst nanoparticles |
| N-$TiO_2$ | Nitrogen doped $TiO_2$ |
| PAN | Polyacrylonitrile |
| PCA | Picrolonic acid |
| PC | Policarbonate |

| | |
|---|---|
| Pd/TiO$_2$ | Palladium doped TiO$_2$ |
| PE | Primary effluent |
| PEBAx | Polyether-polyammide block copolymers |
| PEG | Poly(ethylene glycol) |
| PES | Polyethersulfone |
| PLC | Programmable logic controller |
| PM | Photocatalytic membrane |
| PMR | Photocatalytic membrane reactor |
| PNP | p-nitrophenol |
| PP | Polypropylene |
| PR | Photocatalytic reactor |
| PSF | Polysulfone |
| PTFE | Polytetrafluoroethylene |
| PV | Pervaporation |
| PVDF | Polyvinylidene difluoride |
| PWF | Pure water flux |
| RO29 | Reactive orange 29 |
| RO | Reverse osmosis |
| SE | Secondary effluent |
| SEM | Scanning electron microscopy |
| SMX | Sulfamethoxazole |
| SPMR | Submerged photocatalytic membrane reactor |
| SPMS | Sono-photocatalysis/membrane separation |
| TMP | Transmembrane pressure |
| TOC | Total organic carbon |
| TW | Tap water |
| UF | Ultrafiltration |
| UV | Ultraviolet radiation |
| VA | Vanillin |
| VB | Valence band |
| VIS | Visible radiation |
| WOS | Web of science |
| XRD | X-ray diffraction |

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
