# Peer review of "The Evolution of Photocatalytic Membrane Reactors over the Last 20 Years: A State of the Art Perspective"

_catalysts, doi:10.3390/catal11070775_

Round 1
Reviewer 1 Report
The review covers an interesting topic area that is rapidly growing. The only issue with the paper is the number of grammatical errors and at times sentence structure. In its current form the paper is a little difficult to read. These issues need to be fixed before publishing.
Author Response
Dear Reviewer, thank you for the work done in reviewing our manuscript.
Below, in Italycs, is the answer to your questions.
The review covers an interesting topic area that is rapidly growing. The only issue with the paper is the number of grammatical errors and at times sentence structure. In its current form the paper is a little difficult to read. These issues need to be fixed before publishing.
The English of the manuscript has been reviewed giving a focus on grammatical errors and sentence structures.
Reviewer 2 Report
The authors made a significant work as they have to comprise a large amount of data in this review. The paper can be published only after addressing these issues:
- In the Introduction part the authors must outline why this work is significant for the scientific community, there is something new to learn or to consider in the near future.
- Introduction part should contain a paragraph about the photocatalysts and their composition and morphology (representative work such as DOI: 10.1016/j.matlet.2011.03.111, DOI: 10.1016/j.jcis.2019.09.017. should be included).
- The authors must summaries at the end of each sub-chapter (i.e. PMRs configuration) their own opinion about the pathway to follow.
- The level of criticism must be increased especially in sub-chapter 3.3 and 4.2.
- Conclusion part must be more focus on values and data, as an example of good practice.
- The manuscript require a correction regarding typo and grammar mistakes.
Author Response
Dear Reviewer, thank you for the work done in reviewing our manuscript.
Below, in Italycs, is the answer to your questions.
- In the Introduction part the authors must outline why this work is significant for the scientific community, there is something new to learn or to consider in the near future.
A sentence has been introduced at the end of the introduction (lines 147-148)
2. Introduction part should contain a paragraph about the photocatalysts and their composition and morphology (representative work such as DOI: 10.1016/j.matlet.2011.03.111, DOI: 10.1016/j.jcis.2019.09.017. should be included).
The reference DOI: 10.1016/j.matlet.2011.03.111 has been introduced and now is reference 65.
The reference DOI: 10.1016/j.jcis.2019.09.017 has been introduced and now is reference 70.
A short paragraph has been added in the introduction regarding reference 70 (lines 129-133)
3. The authors must summaries at the end of each sub-chapter (i.e. PMRs configuration) their own opinion about the pathway to follow.
Some considerations have been added at the end of each sub-chapter giving some opinion about the pathway to follow.
4. The level of criticism must be increased especially in sub-chapter 3.3 and 4.2.
Some considerations have been added in these sub-chapters
5. Conclusion part must be more focus on values and data, as an example of good practice.
The conclusion section has been revised.
6. The manuscript requires a correction regarding typo and grammar mistakes.
The English of the manuscript has been reviewed giving a focus on typo and grammatical mistakes.
Reviewer 3 Report
Molinari and co-workers present a review on the research activities on photocatalytic membrane reactors during the last about 20 years. The review is well written and interesting to read. The authors achieve a comprehensive review which is certainly very good for researchers new to the field. The number of citations is appropriate. I, thus, recommend publication of the manuscript in its present form. I only have minor recommendations on the text which will most certainly dealt with during the editing of the galley proofs.
Page 2, line 84, promote(s)
Page 3, line 109/110, “…it should be pointed out that no MS exists which is..”
Page 3, line 119, “…in the last years became..”
Author Response
Dear Reviewer, thank you for the work done in reviewing our manuscript.
Below, in Italycs, is the answer to your questions.
Molinari and co-workers present a review on the research activities on photocatalytic membrane reactors during the last about 20 years. The review is well written and interesting to read. The authors achieve a comprehensive review which is certainly very good for researchers new to the field. The number of citations is appropriate. I, thus, recommend publication of the manuscript in its present form. I only have minor recommendations on the text which will most certainly dealt with during the editing of the galley proofs.
Page 2, line 84, promote(s) DONE
Page 3, line 109/110, “…it should be pointed out that no MS exists which is..” DONE
Page 3, line 119, “…in the last years became..” DONE
Round 2
Reviewer 2 Report
The manuscript can be published in the present form.